

# Traveling/non-traveling phase transition and non-ergodic properties in the random transverse-field Ising model on the Cayley tree

Ankita Chakrabarti[1,2,3*], Cyril Martins[4], Nicolas Laflorencie[3],
Bertrand Georgeot[3], Éric Brunet[5] and Gabriel Lemarié[2,3,6†]

**1** Department of Physics, National University of Singapore,
2 Science Drive 3, Singapore 117542, Singapore
**2** MajuLab, International Joint Research Unit IRL 3654, CNRS, Université Côte d'Azur,
Sorbonne Université, National University of Singapore,
Nanyang Technological University, Singapore
**3** Laboratoire de Physique Théorique, Université de Toulouse, CNRS, UPS, France
**4** Laboratoire de Chimie et Physique Quantiques, Université Paul Sabatier Toulouse III,
CNRS, 118 Route de Narbonne, 31062 Toulouse, France
**5** Laboratoire de Physique de l'École normale supérieure, ENS, Université PSL, CNRS,
Sorbonne Université, Université Paris Cité, F-75005 Paris, France
**6** Centre for Quantum Technologies, National University of Singapore,
117543 Singapore, Singapore

⋆ chakrabarti@irsamc.ups-tlse.fr , † lemarie@irsamc.ups-tlse.fr

## Abstract

We study the random transverse field Ising model on a finite Cayley tree. This enables us to probe key questions arising in other important disordered quantum systems, in particular the Anderson transition and the problem of dirty bosons on the Cayley tree, or the emergence of non-ergodic properties in such systems. We numerically investigate this problem building on the cavity mean-field method complemented by state-of-the art finite-size scaling analysis. Our numerics agree very well with analytical results based on an analogy with the traveling wave problem of a branching random walk in the presence of an absorbing wall. Critical properties and finite-size corrections for the zero-temperature paramagnetic-ferromagnetic transition are studied both for constant (independent of the system volume) and algebraically vanishing (scaling as an inverse power law with the system volume) boundary conditions. In the later case, we reveal a regime which is reminiscent of the non-ergodic delocalized phase observed in other systems, thus shedding some light on critical issues in the context of disordered quantum systems, such as Anderson transitions, the many-body localization or disordered bosons in infinite dimensions.



# 1 Introduction

Characterization of non-ergodic properties has been of utmost importance for studying disorder induced phases of quantum matter ever since Anderson's discovery of the localization of non-interacting electrons in an imperfect crystal [1,2]. It has also emerged as an important issue more recently in the study of isolated quantum many-body systems where both interaction and disorder are present and lead to a new type of phase of matter, the many-body localization (MBL) [3–12]. The localization of all the many-body eigenstates of a system in presence of strong disorder prevents thermalization and has striking consequences on out-of-equilibrium properties [4,13–19].

In this context, there has been a renewed interest on the Anderson transition on random graphs [10,11,20–29,29–46]. One of the major motivations is an approximate mapping between the MBL problem formulated in the Fock space [47–50] and Anderson localization on random graphs [4,6,10–12,51–53]. A topic of much debate has been the existence of a non-ergodic delocalized phase, where states, while not localized as in the strong disordered regimes, only occupy an algebraically small fraction of the system [24–37,40,41,54]. This phase is characterized by multifractal properties. Following its initial discovery, a sub-

stantial body of work has been dedicated to this phase, spanning from mathematical proofs and random matrix models to investigations of Floquet-driven or quasi-periodic systems, see e.g. [55–64]. Note that in finite dimensional system, there is no non-ergodic delocalized phase, and multifractal properties only occur at the Anderson transition [65].

While not random by construction, tree-like structures are likewise good testbed for exploring these ideas. Indeed, classical and quantum systems on the Cayley tree (see Fig. 1) are known to demonstrate strong spatial inhomogeneities in the presence of disorder. Each sites on this tree has $K + 1$ neighbors apart from the leaves at the boundary which have 1 neighbor. The paradigmatic directed polymer model [66] studied in terms of a traveling wave problem [67–69], is known to show a glass transition. In the glassy phase, only a few paths are explored by the polymer from the exponentially many of the tree. For the quantum problem of Anderson localization on the Cayley tree, it was shown that there exists a non-ergodic delocalized phase [27–29, 32]. A noteworthy point is that boundary conditions play a crucial role on the Cayley tree: they must depend on the system size in a certain way for such a non-ergodic phase to exist [27–29, 32]. In many quantum disordered systems an analogy with the directed polymer problem can be drawn [11, 20, 23, 27, 33, 45, 70–74] facilitating the understanding of universal features of the glassy physics [75]. Consequently, such systems with a tree-like structure are particularly illustrative for studying non-ergodic properties of glassy physics.

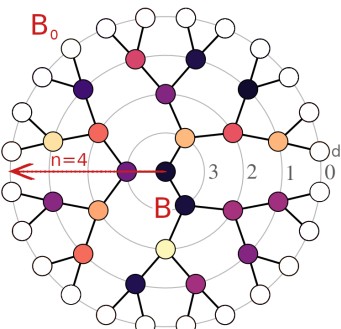

Figure 1: Schematic representation of the Cayley tree. The transverse random field Ising model Eq. (1) lives on a centered Cayley tree of finite depth $n$ and branching number $K$ (on the figure, $n = 4$ and $K = 2$). We label generations by the letter $d$, with $d = 0$ being the boundary and $d = n$ being the site at the center of the tree. The different colors of the nodes for $d > 0$ correspond to a given random configuration of the transverse fields $\epsilon_i$ in the Hamiltonian Eq. (1). The key observable of this work is the cavity mean field $B_d$ at any site within generation $d$. We study the response $B = B_{n-1}$ closest to the center of the tree as a function of a boundary condition $B_0$.

In this paper we study the effects of disorder on a quantum system: the random-field Ising model on the Cayley tree. This model has been widely used to study quantum phase transitions at zero temperature [76] in the presence of frozen disorder. Important methods have been devised to describe its properties, including the strong-disorder renormalization procedure [77, 78] and, in one dimension, the mapping through the Jordan-Wigner transformation to a free fermion problem [79–81]. The first renormalization approach predicts a transition, between an ordered and a paramagnetic phase, of infinite disorder type, where quenched randomness increases as the system undergoes coarse graining [77, 78, 82–84]. The second approach allows for exact numerical simulations of large 1D systems and maps them to the Anderson localization problem [80, 81]. At first glance, it might seem that the absence of loops on the Cayley tree could enable such an approach. However, the Jordan-Wigner transformation there introduces non-trivial and non-local phase terms, which hinder the mapping to a simple free fermion problem [85, 86]. We will use another approach well suited for tree-graphs:

the cavity mean-field (CMF) description of this problem introduced by Feigel'man, Ioffe and Mézard [70], see also [71, 87–89].

An initial motivation to consider this problem was to understand the large spatial inhomogeneities emerging at the superconductor-insulator transition in certain strongly disordered materials where the transition is driven by the localization of Cooper pairs [70, 71, 89, 90]. Such physics can be modelled by disordered hard-core bosons or equivalently $XY$ spin-1/2 systems in the presence of a random magnetic field [91–93]. It was argued in Refs. [70, 71, 89] that some features of this transition can be captured by the random transverse-field Ising model on the Cayley tree, as studied using the CMF approach [70, 71].

In Refs. [70, 71], this problem was mapped to that of directed polymers on the Cayley tree which brought to the fore interesting glassy and Griffiths properties at strong disorder and even in the superconducting phase close to the transition. This provides qualitative explanation for the large spatial fluctuations of the local order parameter observed experimentally in disordered superconducting films close to the transition [89, 90].Moreover, hard-core bosons in a random potential have recently been studied on the Cayley tree [94] where the observed Bose-glass phase [95, 96] was found to share some common features with a delocalized non-ergodic regime.

In this paper, we first aim at investigating in detail the zero-temperature paramagnetic - ferromagnetic quantum phase transition as disorder decreases for the above problem on the Cayley tree. As we will demonstrate, in the cavity mean-field approach, one can draw an analogy between this transition and the traveling/non-traveling phase transition for a branching random walk in the presence of an absorbing wall, as described in [97]. Such an analogy was also suggested for the Anderson transition on the Cayley tree [23, 28, 33]. In fact, recursive equations for the cavity mean fields closely resemble that of Abou-Chacra, Thouless and Anderson for the Anderson localization problem [20], but are simpler in the sense that they involve only real quantities. Interestingly, the analogy between cavity mean field and the traveling/non-traveling phase transition [97] allows us to describe analytically and to verify numerically a range of critical properties for the Ising transition. Some of these properties are identical to those known for the Anderson transition, while others, including critical behavior, have not yet been explored in the context of the Anderson transition.

Furthermore, we will show that the strong similarity between the Anderson transition and that for the random-field Ising model extends to non-ergodic properties, particularly in terms of their crucial dependence on boundary conditions. We primarily focus on the case of a finite Cayley tree, which, in the Anderson model, allows for a clear characterization of a non-ergodic delocalized phase [27–29, 32]. When considering boundary conditions in the Ising problem analogous to those in the Anderson localization problem that are suitable for non-ergodic properties, we identify signs of a "non-ergodic ordered phase", that we will define precisely in the following.

The rest of the paper is organized as follows. In Sec. 2 we recall the context of the study and its main objectives. In Sec. 3 we present the model and the CMF method. In Sec. 4 we draw an analogy with a traveling wave problem with a wall considered in [69, 97] in order to reach a better understanding of the critical properties and the conditions of existence of the equivalent of a non-ergodic delocalized phase in this context, that we will call non-ergodic ordered phase in the following. We first recall the analogy in the disordered regime between the cavity mean field equation with the directed polymer problem [70]. In turn, this problem maps onto a traveling wave problem. Close to the transition and in the ordered phase, the nonlinearity of the recursion cannot be neglected and the problem maps instead to a traveling wave problem with a wall, which was considered in [69, 97]. Numerical results corresponding to a uniform boundary field are presented. Behaviors in the traveling and stationary regimes are compared with the analytical predictions made. Then a finite-size scaling study of these

results is performed and compared with the critical behavior predicted theoretically. In Sec. 5 we show that this system exhibits a non-ergodic ordered phase for an appropriate choice of the boundary conditions. This phase is studied in the light of the analogy with the traveling wave problem. The strongly inhomogeneous spatial distributions of cavity fields observed in different phases is discussed in Sec. 6. We summarize our results in Sec. 7.

## 2 Main objectives

In this paper, we study the ordered/disordered phase transition in a spin system, the random transverse-field Ising model on the Cayley tree, addressing two important questions which were recently debated in the delocalized/localized Anderson transition: (i) What are the critical properties of the transition, in particular how to describe the finite-size scaling properties in the neighborhood of the transition? (ii) Is there a phase in such a spin system which is analogous to the non-ergodic delocalized phase of the Anderson problem, and what are the conditions for its existence?

In the first question, we consider the system on a Cayley tree with fixed boundary conditions. In this setting, there is no non-ergodic ordered phase (analogous to the non-ergodic delocalized phase in Anderson), and we wish to study the critical properties of the direct disordered to ergodic ordered phase transition, akin to the direct localized to ergodic delocalized AL transition. The first question is important because of a recent debate on the value of critical exponents of the Anderson localization transition [29, 29, 32, 33, 36, 41, 98]. In the CMF problem we consider, we will draw an analogy with a traveling/non-traveling phase transition whose critical properties (in particular the critical exponents) are analytically known. This will allow us to perform a finite-size scaling of this problem which could in turn be useful to analyse the Anderson transition on the Cayley tree.

The second question of whether a non-ergodic ordered phase exists for the quantum Ising model that we consider, similar to the non-ergodic delocalized phase of the Anderson localization problem, is also crucial. Indeed, the recent study of disordered bosons on the Cayley tree [94] points towards the existence of a Bose-glass phase with properties very similar to those of a delocalized non-ergodic phase, and also showing some multifractality. In the recursive approach for the Green's function of the Anderson localization problem on tree graphs, it was understood that the condition for observing such a non-ergodic delocalized phase is that the boundary conditions must tend to zero as the inverse of the number of sites [27,29,32,99]. Indeed, in the Cayley tree, the boundary hosts a finite fraction of the total number of sites, and therefore plays a preponderant role. In this paper, we will show that by considering boundary conditions vanishing as a power-law $\sim 1/N^\phi$, where $N$ is the total number of sites, we can also observe the equivalent of a non-ergodic delocalized phase, and an ergodicity breaking transition.

## 3 Method

We study the random transverse-field Ising model described by the following Hamiltonian

$$H = -\sum_i \epsilon^{(i)} \sigma_i^z - \frac{g}{K} \sum_{\langle ij \rangle} \sigma_i^x \sigma_j^x, \tag{1}$$

where $\sigma^z$, $\sigma^x$ are the Pauli matrices. The lattice considered is the Cayley tree with a branching number $K$. The summation in the second term couples the spin at each site $i$ to that of the $K + 1$ nearest neighbours. The transverse fields $\epsilon^{(i)}$ at each site $i$ are independent random

variables uniform in $[-1, 1]$; in other words, they are drawn from a box distribution,

$$P(\epsilon) = \frac{1}{2}\theta(1 - |\epsilon|). \tag{2}$$

We aim at analyzing the properties of the paramagnetic-ferromagnetic transition at zero temperature, obtained by decreasing the relative strength of the disorder. This is achieved by varying the coupling $g$ from small to large values. The phase diagram of this transition for the infinite, boundaryless, Cayley tree (also called the Bethe lattice), has been investigated using a CMF approach [70, 71] where it was found that the disordered phase at sufficiently low temperature has characteristic glassy properties, in particular replica symmetry breaking and large distributions of the local order parameter, which persist in the ordered phase close to the transition.

In the CMF approach, we consider a spin at site $j$ from which the neighbor closer to the root has been removed (cavity). It is described by a local Hamiltonian:

$$H_j^{\text{CMF}} = \epsilon^{(j)}\sigma_j^z - \frac{g}{K}\sum_{l=1}^{K}\langle\sigma_l^x\rangle\sigma_j^x, \tag{3}$$

where $\langle\sigma_l^x\rangle$ is the magnetization at the site $l$ obtained (recursively) from the CMF approach itself: to compute $\langle\sigma_l^x\rangle$, we exclude the site $j$ and only consider the sites further away from the center. In a finite Cayley tree of depth $n$, it is useful to introduce the distance $d$ from the boundary: it increases from 0 to $n$ as we move inwards from boundary to root, see Fig. 1. Assume site $j$ is at level $d + 1$; then all the sites $l$ in Eq. (3) are at level $d$. We introduce the cavity mean field for $j$ as

$$B_d^{(j)} = \frac{g}{K}\sum_{l=1}^{K}\langle\sigma_l^x\rangle, \tag{4}$$

where the index $d$ recalls that this quantity is an average of magnetizations of some sites at level $d$. At zero temperature, solving Eq. (3), one finds $\langle\sigma_j^x\rangle = B_d^{(j)}/\left(\epsilon^{(j)2} + B_d^{(j)2}\right)^{1/2}$. Then, the cavity mean field of a site $j'$ at level $d + 2$ is given by

$$B_{d+1} = \frac{g}{K}\sum_{l=1}^{K}\frac{B_d^{(l)}}{\sqrt{\epsilon^{(l)2} + B_d^{(l)2}}}, \tag{5}$$

where we have dropped the $(j')$ superscript of the left hand side to shorten notation. The quantities $B_d^{(l)}$ are random variables with a distribution depending only on the index $d$. For a given site $l$ at level $d + 1$, the quantities $B_d^{(l)}$ and $\epsilon^{(l)}$ are independent. For a given $d$, the quantities $B_d^{(l)}$ and $B_d^{(m)}$ corresponding to two different sites $l$ and $m$ are independent.

In this paper, we will mainly consider the response of the system at the root of the tree to a uniform and non-random magnetization $\langle\sigma_l^x\rangle = B_0/g$ for all the sites $l$ at the boundary $d = 0$: this corresponds to having cavity field $B_0$ for all the sites at level $d = 1$. We use Eq. (5) for each of the preceding generation up to the root to obtain the field $B_d^{(j)}$ for all $j$ and $d$. We analyze the critical properties by studying the distribution of the cavity field at the root, for $d = n - 1$. (Remark: at the root, the cavity mean field is still defined by Eq. (4), with the sum running over an arbitrary choice of $K$ out of $K + 1$ sites at level $n - 1$.) For simplicity, we write henceforth

$$B = B_{d=n-1}^{(\text{root})}. \tag{6}$$

The distribution of the order parameter $B$ depends on the coupling $g$, the depth of the tree $n$ and the boundary field $B_0$. We will consider a branching number $K = 2$. The notations are illustrated in Fig. 1.

When the depth $n$ of the Cayley tree goes to infinity, we expect the distribution of $B_d$ to converge, as $d \to \infty$, to some stationary distribution $P(B)$. As the random variables in the right hand side of the recursion relation Eq. (5) are all independent from each other, we see that Eq. (5) determines a self-consistent equation for the distribution $P(B)$. For numerical studies, approximate methods like the belief propagation method [23,32] are used to find solutions for the distribution $P(B)$. This approach is however accurate only in the bulk of a large Cayley tree where the boundary can be ignored, as previously considered in [70,71] for the random transverse field Ising model.

We consider in this paper the case of a finite Cayley tree of depth $n$ for two main reasons. First because we want to investigate the finite-size scaling properties of this transition to extract its critical properties. Finite-size scaling in the belief propagation method approach is not clearly understood (see however [23,33]). Critical exponents are then determined from the asymptotic behavior (by definition far from the critical regime) of the considered observable, see [36]. Although solving exactly CMF recursive equation Eq. (5) for a tree of finite size restricts the depth of the tree quite strongly (up to $n = 29$ in our case), we will see that finite-size scaling allows to determine precisely the critical exponents of the transition. The second reason is that the finite Cayley tree case shows strong and interesting boundary effects. Indeed, the sites at the boundary have a connectivity 1 instead of $K + 1$, which makes them qualitatively different. Moreover, the number of sites on the boundary is a significant fraction $1 - K^{-1}$ for large trees of the total number of sites $N = (K+1)(K^n - 1)/(K-1) + 1$. Similarly to the Anderson transition on the Cayley tree, we will see that boundary conditions control the properties of the ordered phase, and that a non-ergodic regime can appear in some cases.

The CMF approach is in fact similar to that used to describe Anderson localization in the Cayley tree [20,21,23,25,26,29,32,36,38,41,72], where we consider the local cavity Green's function $G_{ii}^{(j)}$ of the problem where the site $j$, neighbor of $i$ closer to the root, has been removed. It can be rigorously shown for the Cayley tree that this observable satisfies a recursion equation similar to Eq. (5). The crucial quantity is the imaginary part of the local Green's function $\Im G$ at the root of the tree in response to a boundary condition $\eta$ (the imaginary part of the cavity Green's function at the boundary sites). If $\Im G$ reaches, in the limit of large system sizes, a finite value (i.e. almost surely non-zero), then the system is in a delocalized regime. It was understood recently [25–30,32] that the non-ergodic delocalized phase of the Anderson transition can only be seen by considering a boundary condition $\eta \sim N^{-\phi}$, which vanishes as a power law of the system volume with $\phi$ a constant exponent. In that latter case, the typical value of $\Im G \sim N^{\mathcal{D}-1}$ varies as a power-law of the system volume with a disorder-dependent exponent $0 < \mathcal{D} < 1$. $\mathcal{D}$ can be interpreted as a fractal dimension, and $\mathcal{D} = 1$ in the ergodic regime while it reaches 0 at the localization transition.

We wish here to revisit the problem of the ferromagnetic-paramagnetic transition in the random transverse-field Ising model Eq. (1) in the light of this discovery. In our problem, $B$ corresponds to $\Im G$ and the boundary condition $B_0$ to $\eta$. We will therefore see that here too we can have the equivalent of a non-ergodic delocalized phase if the boundary condition satisfies $B_0 \sim N^{-\phi}$.

## 4 Traveling/non-traveling wave phase transition

In this section, we discuss the analogy with the traveling/non-traveling wave phase transition of Derrida and Simon [97]. We consider the case where the value of the boundary field $B_0$ is independent of the size $N$ of the system. (We will investigate the case of $B_0 \sim N^{-\phi}$ coresponding to the non-ergodic delocalized phase in the Anderson transition on the Cayley tree in Sec. 5.)

We argue that, when $B_0$ is independent of $N$, the CMF recursion Eq. (5) describes a transition between a disordered phase for $g < g_c$ where the field $B = B_{n-1}$ goes to zero as $N \to \infty$, and an ergodic ordered phase for $g > g_c$, where the field converges to some fixed distribution as $N \to \infty$.

We will first describe analytically the critical properties of this transition. Then, we will verify these predictions by analysing the numerical results obtained for a uniform boundary field, $B_0 = g$. Note that this choice corresponds to the upper bound of the cavity field $B \le g$. We have considered, unless stated, $N_s = 10^4$ independent realizations of the disorder for each value of $n$ and $g$.

## 4.1 Traveling wave predictions for the disordered phase

In the disordered phase $g < g_c$, because $B$ vanishes, we can linearize the recursion relation Eq. (5), which then gives:

$$B_{d+1} = \frac{g}{K} \sum_{l=1}^{K} \frac{B_d^{(l)}}{|\epsilon^{(l)}|} . \tag{7}$$

This recursion is similar to that of a directed polymer problem on a Cayley tree [66], a paradigmatic model of pinned elastic manifolds with characteristic glassy properties, in particular 1-step replica symmetry breaking. More precisely, Eq. (7) is equivalent to the recursion relation found for the partition function of a directed polymer:

$$Z_{d+1} = \sum_{l=1}^{K} e^{-\beta U^{(l)}} Z_d^{(l)} , \tag{8}$$

where the partition function $Z$ plays the role of $B$, the inverse temperature $\beta$ is arbitrarily set to 1 and the on-site disorder is $U^{(l)} \equiv \ln(K\epsilon^{(l)}/g)$. In turn, the problem of directed polymer on the Cayley tree can be mapped to a traveling wave problem [66].

This analogy allows us to deduce a number of analytical results, which are well known in the context of the traveling wave problem but new for the problem considered here. We explain these results in more details and give some references in Appendix A. When Eq. (7) holds, then

$$\ln B = -V(g)n - \frac{3}{2\gamma} \ln n + \ln B_0 + \ln b_n , \tag{9}$$

where $b_n$ is a random variable of order 1 and where $\gamma$ and $V(g)$ are given below. Furthermore, the random variable $b_n$ converges (in distribution) as $n \to \infty$ to some random variable $b$

$$b_n \xrightarrow[n \to \infty]{} b , \tag{10}$$

and the density of probability of $b$ has a fat tail for large $b$:

$$P(b) \simeq (C_1 \log b + C_2) b^{-(1+\gamma)} , \quad \text{for large } b , \tag{11}$$

for some constants $C_1$ and $C_2$. Finally, the velocity $V(g)$ can be written as

$$V(g) = \ln \frac{g_c}{g} , \tag{12}$$

for some critical value $g_c$. The values of $g_c$, $\gamma$ and the full distribution of $b$ depend on the choice of $K$ and on the distribution of the noise $\epsilon$. When $K = 2$ and for $\epsilon$ uniform in $[-1, 1]$, as in Eq. (2), one finds (see Eq. (A.12) and Eq. (A.13) with $\gamma \equiv \lambda_c$ and $V(g) \equiv -v_c$, and [70])

$$g_c \simeq 0.137353 , \qquad \gamma \simeq 0.626635 . \tag{13}$$

Notice in particular that the exponent $\gamma$ in the disordered phase is independent of the value of $g$.

Fat tails as given by Eq. (11) are a typical feature of a non-ergodic phase [23, 66, 70, 100]. The above value ($\gamma < 1$) corresponds to the replica symmetry broken glassy phase of the auxiliary directed polymer problem [66, 70], where the average value is dominated by the rare events. $\gamma$ also controls the logarithmic corrections to the linear decay of $\ln B$ with $n$. As we will show, these corrections are particularly important at small $n < 30$, as considered in this work.

## 4.2 Numerical evidence for the traveling wave regime

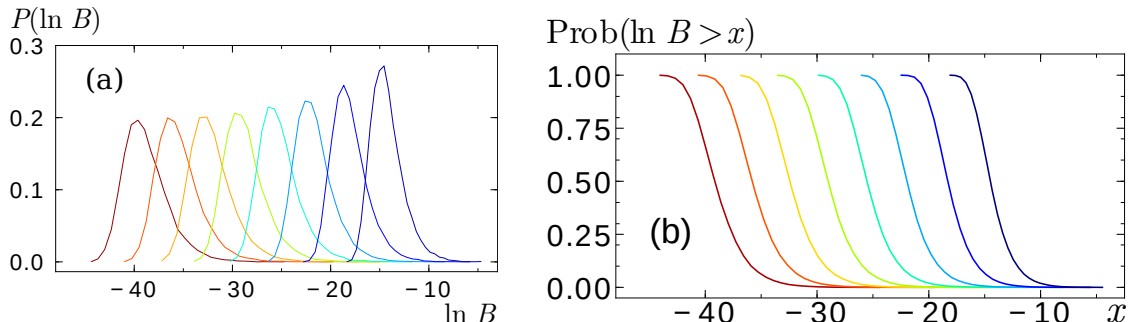

Figure 2: (a) Distributions of $\ln B$ deep in the disordered phase, $g = 0.03 < g_c$, for trees with total number of generations $n = 6, 8, 10, \cdots, 20$, from right to left. $P(\ln B)$ is a traveling wave propagating to the left with the increase of $n$, preserving its shape (for $n \geq 14$) with increasing $n$. (b) Corresponding cumulative distribution functions $\mathrm{Prob}(\ln B > x)$.

In the disordered paramagnetic phase for $g < g_c$, Eq. (9) predicts a traveling wave moving with a velocity $V(g)$. In Fig. 2 we have plotted the distributions $P(\ln B)$ and the cumulative distribution functions $\mathrm{Prob}(\ln B > x)$ for trees with total number of generations $n = 6 - 20$, deep in the disordered phase, $g = 0.03$. With the increase in $n$ the distributions move to the left to progressively smaller values preserving their shape (for sufficiently large trees), which well reflects the traveling wave solution.

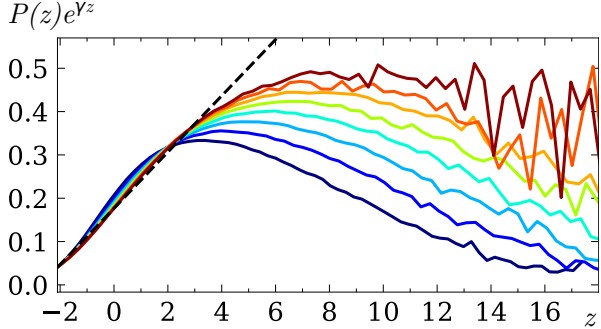

Figure 3: Plot of $P(z)e^{\gamma z}$ as a function of $z$, where $z := \ln B - \langle \ln B \rangle$, for $g = 0.03$ (deep in the disordered phase) and system sizes $n = 10, 12, \ldots, 24$ with $n = 10$ at the bottom and $n = 24$ at the top. The number of disorder configurations considered are $10^7$ for system sizes $n \leq 20$ and $10^6$ for $n = 22$ and 24. From Eq. (9), one has $z = \ln b_n - \langle \ln b_n \rangle$. From Eq. (11), one expects for fixed large $z$ to have $P(z) \simeq (C_1' z + C_2')e^{-\gamma z}$ as $n \to \infty$. We observe indeed that the curves for $P(z)e^{\gamma z}$ for $z > 0$ collapse onto a straight line (indicated by a dashed line) as $n$ increases.

We now focus on the centered shape of the distribution, *i.e.* on the distribution of $z = \ln B - \langle \ln B \rangle$. First, notice from Eq. (9) that $z$ can also be written as $z = \log b_n - \langle \log b_n \rangle$. Then, the prediction Eq. (10) means that the distribution of $z$ converges as $n$ becomes large to some limiting distribution. This is consistent with the observation in Fig. 2 that for large $n$, the shape of the distribution of $\ln B$ relative to its peak converges. Furthermore, the prediction Eq. (11) implies that, for a fixed large $z$, the distribution of $z$ should satisfy, in the $n \to \infty$ limit, the following behaviour: $P(z) \simeq (C_1' z + C_2') e^{-\gamma z}$. We check that prediction in Fig. 3 where we plot $P(z) e^{\gamma z}$ as a function of $z$ for system sizes $n = 10$–$24$. As $n$ increases the curves for $z > 0$ collapse to a straight line corresponding to the predicted behaviour $C_1' z + C_2'$.

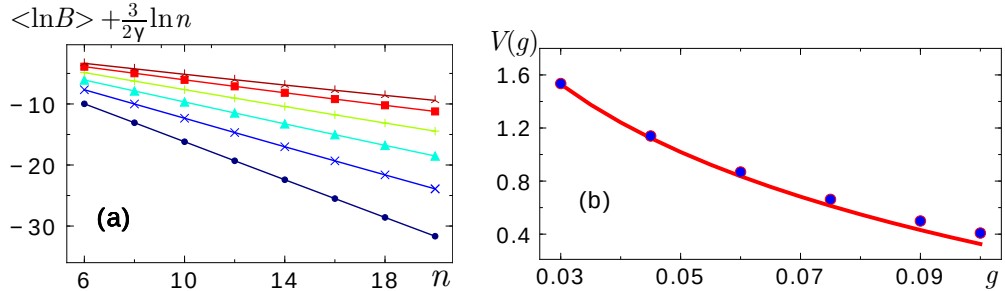

Figure 4: (a) Decay of $\langle \ln B \rangle$ with system size $n$ in the disordered phase, for $g$ taking the value $g = 0.03, 0.045, 0.06, 0.075, 0.09$ and $0.1$ from bottom to top ($g_c \approx 0.137$). We plot $\langle \ln B \rangle + \frac{3}{2\gamma} \ln n$ with $\gamma \approx 0.627$ (see text and Eq. (9)) and the corresponding linear fits (shown by solid lines) give the velocity $V$. (b) Behavior of the velocity $V$ as a function of $g$ as compared to the theoretical prediction Eq. (12) (red curve) in the linear approximation. Fitted $V$ values are in good agreement with Eq. (12) for $g$ far enough from the transition, where the linearization is a good approximation for the range of $n$ considered.

In Fig. 4 we test numerically another prediction for the disordered behavior, Eq. (9), describing the linear decay of $\langle \ln B \rangle$ with $n$ and Eq. (12) giving the $g$-dependence of the velocity in the linear approximation. Numerical data in the disordered regime $g \leq 0.1$, for system sizes limited to $n \leq 20$, clearly show that logarithmic corrections described by Eq. (9) are important. Taking them into account, the determined velocity $V$ is in good agreement with the prediction Eq. (12), for $g$ far enough from the transition where the linearization is a good approximation for the range of sizes considered. Near the transition, the deviations observed are probably due to finite size effects.

## 4.3 Stationary distribution for the ordered phase

Notice in Eq. (9) that $V(g) > 0$ and $B \to 0$ if $g < g_c$, whereas $V(g) < 0$ and $B \to \infty$ as $g > g_c$. We claim that this same value of $g_c$ also controls the transition between the ordered and disordered phase in the original model Eq. (5):

- if $g < g_c$, then $B$ as obtained from Eq. (5) goes to zero with the system size in the way described by Eq. (9); the system is in the disordered phase.

- if $g > g_c$, then $B$ as obtained from Eq. (5) cannot go to zero and Eq. (9) no longer holds because the non-linearity of the recursion Eq. (5) can no longer be neglected. As it is clear from Eq. (5) that $B \leq g$, the only possibility is that $B$ reaches a stationary distribution in the limit of large sizes; the system is in the ordered phase.

We therefore have a phase transition between a traveling and a non-traveling regimes. As detailed in the appendix, we believe [23, 33] this transition to be similar to the one for a

branching random walk in presence of an absorbing wall described in [97]. We now simply list the predictions explained in detail in the appendix.

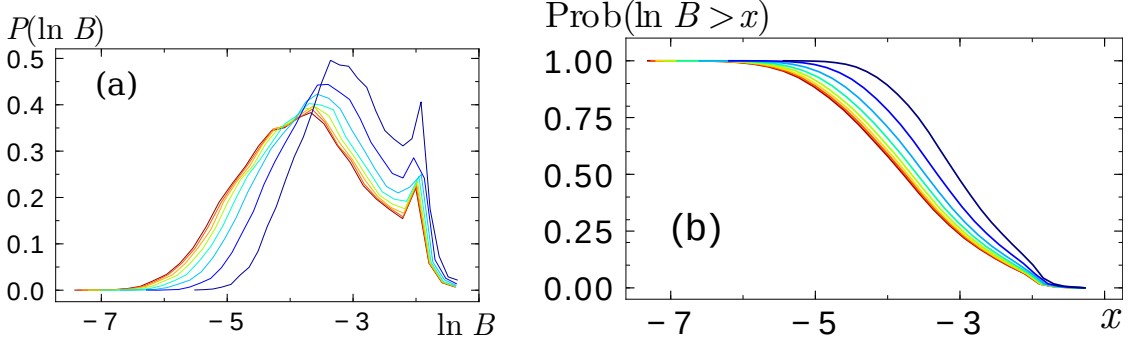

Figure 5: (a) Distributions of $\ln B$ deep in the ordered regime, $g = 0.28$ for trees with total number of generations $n = 6, 8, 10, \cdots, 20$, from right to left. The distributions have converged to a stationary one for larger trees. The origin of the apparent bimodal shape of the distribution has been discussed in [89]. (b) Corresponding cumulative distribution functions $\text{Prob}(\ln B > x)$.

In the ordered phase $g > g_c$, the distribution of $B$ converges to some limiting distribution as the system size increases, as shown in Fig. 5 for $g = 0.28$. Moreover, when $g$ is close to $g_c$, the average value of $\ln B$ is a large negative number described by a critical exponent $\kappa$

$$\langle \ln B \rangle \simeq -C(g - g_c)^{-\kappa}, \tag{14}$$

for $g > g_c$, $g$ close to $g_c$ and $n \to \infty$. Moreover, the distribution of $\ln B$ is concentrated around its expected value, with exponential tails.

As explained in subsection A.6 of the appendix, the analogy with travelling waves [97] suggests that $\kappa = 1/2$, and a prediction for the value of $C$ is given in Eq. (A.31), with $L$ being $-\langle \ln B \rangle$. Furthermore, the tail of the distribution of $\ln B$ is predicted to be (see Eq. (A.33) with $a = L$)

$$\text{Prob}(\ln B > -L + x) \simeq \hat{C} L \sin \frac{\pi x}{L} e^{-\gamma x}, \tag{15}$$

in the region where both $x$ and $L - x$ are large, where $L = -\langle \ln B \rangle$ is given in Eq. (14) and $\hat{C}$ is some constant.

## 4.4 Critical properties

At the transition $g = g_c$, one expects the average value of $\ln B$ to vanish with the system size with another critical exponent $\delta$:

$$\langle \ln B \rangle \simeq -C' n^{\delta}, \tag{16}$$

i.e. $B^{\text{typ}} \equiv \exp(\langle \ln B \rangle)$ vanishes with $n$ as a stretch exponential. As explained in subsection A.7 of the appendix, the analogy with [97] suggests that $\delta = 1/3$, and gives a prediction for $C'$, see Eq. (A.36) with $d = n$ and $L_d = -\langle \ln B \rangle$.

A particularly interesting property is the tail of the distribution of $\ln B$ at criticality $g = g_c$ [69, 97]: it is still given by Eq. (15), but now with $L = -\langle \ln B \rangle \simeq C' n^{\delta}$, see Eq. (16).

There is another, trivial, critical exponent in Eq. (12); if $g < g_c$ but $g$ close to $g_c$, one has

$$V(g) \simeq \frac{1}{g_c}(g_c - g), \quad g < g_c. \tag{17}$$

Finally, we expect $\langle \ln B \rangle$ to follow a single parameter scaling function in the form:

$$\langle \ln B \rangle = -n^{\delta} \mathcal{F}[n^{1/\nu}(g - g_c)]. \tag{18}$$

The behavior of the scaling function $\mathcal{F}$ is such that one should recover the disordered behavior, Eq. (9) and Eq. (17) when $g < g_c$; this leads to

$$\mathcal{F}(X) \sim X^{\nu(1-\delta)}, \quad \text{for } X \to -\infty, \tag{19}$$

with the constraint:

$$1 = \nu(1-\delta). \tag{20}$$

With the value $\delta = 1/3$ predicted for the traveling/non-traveling phase transition in [97], we obtain $\nu = 3/2$. Moreover, Eq. (19) should allow to recover the behaviour Eq. (14) of the ordered phase, in the asymptotic limit $n \gg |g - g_c|^{-\nu}$, which implies

$$\mathcal{F}(X) \sim X^{-\nu\delta}, \quad \text{for } X \to +\infty, \tag{21}$$

and, therefore, the following relation between the critical exponents:

$$\kappa = \nu\delta. \tag{22}$$

With the values of the critical exponents $\nu = 3/2$ and $\delta = 1/3$ predicted for the traveling/non-traveling phase transition in [97], we obtain $\kappa = 1/2$.

## 4.5 Numerical characterization of the critical behavior

In Figs. 6 and 7 we test numerically the theoretical predictions for the critical behavior at $g = 0.137 \approx g_c$. Fig. 6 (a) first shows the power law decay of $\langle \ln B(g_c) \rangle$, as in Eq. (16). Incorporating small system size corrections, we find that the critical data are compatible with a value of the critical exponent $\delta = 1/3$. More precisely, we fit the data with the following function:

$$\langle \ln B(g_c) \rangle = -C'(n - c_1)^{1/3} + c_2, \tag{23}$$

with the theoretical value $C' \approx 5.534$ given by Eq. (A.36) and two fitting parameters: $c_1 \approx 2.1$ and $c_2 \approx 3.0$.

In Figs. 6 (b) and 7 we verify the form Eq. (15) of the cumulative distribution with $L$ replaced by a $n$-dependent quantity of the order of $-\langle \ln B(g_c) \rangle$ at criticality. We first plot $\text{Prob}(\ln B > x)e^{\gamma x}$ as a function of $x$ for different system sizes in Fig. 6(b). These curves have the form of an arch. They all vanish at $x \approx -D = -2.65$ and their maximal value $M(n)$ is reached at $x = -D - L(n)/2$, thus defining $M(n)$ and $L(n)$. When plotted as a function of the scaled variable $\pi(x + D)/L(n)$, the different arches corresponding to $\text{Prob}(\ln B > x)e^{\gamma x}/M(n)$ all collapse onto a single curve, as expected from Eq. (15). This is shown in Fig.7(a).

The prediction from the appendix is that, for large system sizes, one should have $L(n) \sim -\langle \ln B \rangle$. Then, the prediction Eq. (15) gives a maximum for $\text{Prob}(\ln B > x)e^{\gamma x}$ which is asymptotically $M(n) \propto L(n)e^{-\gamma L(n)}$. In Fig. 7 (b) $L(n)$ is shown as a function of $-\langle \ln B(g_c) \rangle$. The observed linear behavior with a slope close to 1 confirms the prediction of Eq. (15). In Fig. 7(c), $M(n)e^{\gamma L(n)}$ is plotted as a function $L(n)$. We again find a linear behavior, in agreement with the theoretical prediction.

All of these numerical results give a clear confirmation of the non-trivial predictions from the analogy with the traveling–non-traveling transition described in [97]. In particular, they confirm the compatibility of our results with the critical exponent $\delta = 1/3$.

## 4.6 Finite size scaling of the critical properties

We have inspected the above transition closely by performing a scaling analysis of the dependence of $\langle \ln B \rangle$ on the size of the trees, in the vicinity of the transition. A key point of the

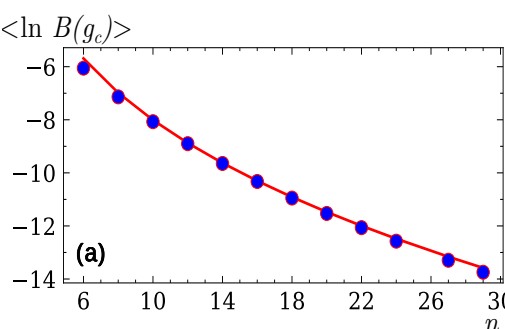

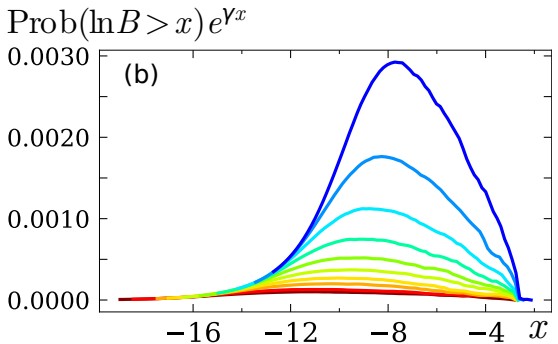

Figure 6: Critical behavior at $g_c = 0.137$. (a) $\langle \ln B(g_c) \rangle$ is well described by Eq. (16), with the predicted critical exponent $\delta = 1/3$, incorporating small system size corrections described by Eq. (23) with two fitting parameters $c_1 \approx 2.1$ and $c_2 \approx 3.0$. The corresponding fit is shown by the red curve. (b) The cumulative distribution $\mathrm{Prob}(\ln B > x)$ is studied for different system sizes $n = 6, 8, 10, 12, 14, 16, 18, 20, 22, 24, 27$ and 29. As expected, when plotted as $\mathrm{Prob}(\ln B > x)e^{\gamma x}$, we obtain a sinus arch which broadens and whose amplitude decreases as a function of system size ($n = 29$ corresponds to the sinus arch at the bottom and $n = 6$ to the topmost). This follows closely the behavior predicted by Eq. (15) with $L$ replaced by a $n$-dependent quantity of the order of $-\langle \ln B \rangle$, as shown in Fig. 7.

considered transition is that we know the exact value of $g_c$: $g_c \approx 0.137353$, Eq. (13). Generally, this information is not known and constitutes a free parameter in the scaling analysis which can lead to great uncertainty on the critical behaviors, in particular on the values of the critical exponents. We have checked the validity of the one-parameter scaling hypothesis Eq. (18) by fitting our numerical data with the following law:

$$\langle \ln B \rangle - \ln B_0 = \tilde{c}_2 - C' m^\delta F(\Delta g \, m^{1/\nu}), \qquad (24)$$

$$m = n - \tilde{c}_1, \qquad (25)$$

with $\Delta g = (g - g_c) + A_2(g - g_c)^2 + A_3(g - g_c)^3$ and $F(X) = \sum_{k=0}^{3} E_k X^k$. $C' = 5.534$, $\delta = 1/3$, $g_c = 0.137353$ are fixed, while $\nu$, $\tilde{c}_1$, $\tilde{c}_2$, $A_2$, $A_3$ and $E_k$, $k = 0, \cdots, 3$, are fitting parameters. The form of the scaling function incorporates the irrelevant small size corrections that we have found for the critical behavior, see Eq. (23) and Fig. 6. We have also subtracted to $\langle \ln B \rangle$ its boundary value $\ln B_0 = \ln g$ whose dependence on $g$ is not related to the critical properties. We have chosen the minimal orders of the Taylor expansions of $\Delta g$ and $F$ which give a non-zero goodness of fit.

The result of such a fitting procedure is the following. We are able to find a fit compatible with the data, within error bars, if we restrict to $n \geq 8$ (our data go up to $n = 27$ and $g$ values are $0.1 \leq g \leq 0.17$. This is quantitatively assessed by the goodness of fit $Q = 0.73$ (see [101]). We find $\nu \approx 1.4$ quite close to the theoretical value $\nu = 3/2$. $E_0 \approx 1.1$ also close to the expected value 1 corresponding to Eq. (23). In Fig. 8, we have plotted the quantity

$$\Lambda \equiv \frac{\langle \ln B \rangle - \langle \ln B_0 \rangle - \tilde{c}_2}{C' m^\delta}, \qquad (26)$$

as a function of $m/\xi$ with $\xi \equiv |\Delta g|^{-1/\nu}$. The data are seen to collapse nicely onto a single scaling function with two branches, the lower one corresponding to the disordered regime $g < g_c$ and the upper one to the ordered regime $g > g_c$. The fitted scaling function $F$ is shown by the dark green and magenta lines. This confirms the validity of the single parameter scaling hypothesis Eq. (18) and the theoretical value of the critical exponent $\nu = 3/2$.

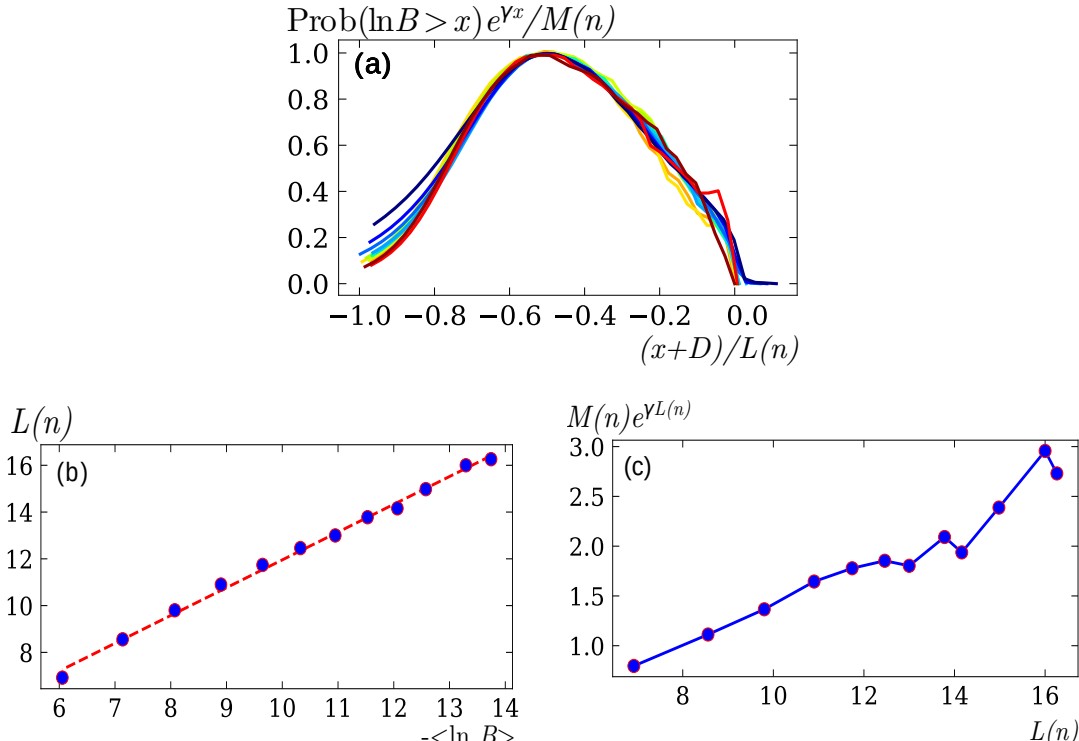

Figure 7: Test of the theoretical prediction Eq. (15) for the critical behavior of the cumulative distribution Prob($\ln B > x$). (a) The distributions for the system sizes $n = 6 - 24$ (increased in steps of 2), 27 and 29 shown in Fig. 6(b) all collapse onto a single shape looking like a sinus arch when Prob($\ln B > x$)$e^{\gamma x}/M(n)$ is plotted as a function of $\pi(x + D)/L(n)$. The scaling parameters $M(n)$, $L(n)$ and the constant $D$ have been determined as follows. $D$ corresponds to the upper vanishing of all the curves for Prob($\ln B > x$)$e^{\gamma x}$ at $x \approx -D = -2.65$. $M(n)$ corresponds to the maximal value of these curves. This maximum is attained at $x = -D - L(n)/2$, which in turn defines $L(n)$. (b) $L(n)$ is shown as a function of $-\langle \ln B(g_c) \rangle$. The behavior is clearly linear $L \approx -1.19\langle \ln B \rangle + 0.07$ with a slope close to 1 (shown by the dashed line), which confirms the theoretical expectation Eq. (15). (c) $M(n)e^{\gamma L(n)}$ is plotted as a function of $L(n)$. As expected from Eq. (15) we observe a linear behavior.

# 5 Non-ergodic phase

Investigation of the existence of a non-ergodic delocalized phase has been a topic of active research in the context of both Anderson localization and MBL [24–29, 32, 54, 94]. In the non-ergodic delocalized phase of the Anderson transition, the states are delocalized within a large number of sites $\propto N^{\mathcal{D}}$ where $0 < \mathcal{D} < 1$; however these sites represent only an algebraically small fraction $\propto N^{\mathcal{D}-1}$ of the volume of the system. $\mathcal{D}$ is called the fractal dimension of the support of the states. For the Anderson transition in finite dimension, such behavior is observed only at the transition threshold, where the states are multifractal [65]. However, it has recently been realized that a number of models, in particular the Anderson problem on the Cayley tree [27–29, 32], display an *extended phase* having these properties: $0 < \mathcal{D} < 1$ in a finite range of disorder strengths $W_1 < W < W_2$. Similar features were found recently in the case of dirty bosons on the Cayley tree [94], a problem closely related to the system we consider here. Taking inspiration from these studies, in this section we explore the possibility of observing such non-ergodic properties in the ordered phase $g > g_c$.

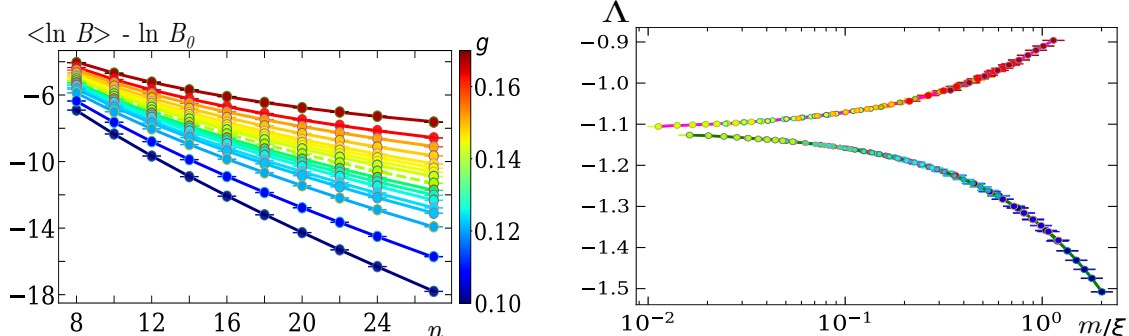

Figure 8: Finite-size scaling analysis of the traveling/non-traveling transition. (a) Raw data: $\langle \ln B \rangle - \ln B_0$ as a function of $n$ in the vicinity of the transition, $g = 0.1, 0.11, 0.12, 0.125, 0.127, 0.13, 0.132, 0.134, 0.137, 0.14,$ $0.142, 0.144, 0.146, 0.15, 0.155, 0.16, 0.17$ from below to top. The data are shown by dots with their error bar, while the lines represent the global fit by the scaling function Eq. (24). The pale green dots (data) and dashed curve (fit) correspond to the critical behavior $g = 0.137 \approx g_c$. (b) Scaling function: The data in (a) are fitted by a single parameter scaling function given by Eq. (24) with the critical value $g_c = 0.137353$ and the critical exponent $\delta = 1/3$ fixed. The best fit gives a value of $\nu \approx 1.4$ close to the theoretical expectation $\nu = 3/2$ for the critical exponent of the analogous traveling–non-traveling transition [97]. When $\Lambda$ given by Eq. (26) is plotted as a function of $m/\xi$ with $m = n - c_1$ and $\xi \sim |g - g_c|^{-\nu}$ (see text), the data collapse onto a single scaling function. The fitted scaling function $F$ is shown by the dark green and magenta lines. This confirms the predictions for the single paramater scaling law Eq. (18).

We observed in Sec. 3 that a major fraction of the total number of sites lie on the boundary of the Cayley tree. Hence, to study the non-ergodic delocalized phase of the Anderson transition on the Cayley tree, it was found that one needs to consider a boundary condition for the imaginary part $\eta$ of the Green's function which vanishes as $N^{-\phi}$ [27–29,32] with $\phi$ a constant exponent. $\eta$ corresponds to the boundary field $B_0$ here, so that we need to appropriately scale the boundary field and choose $B_0 \propto N^{-\phi}$.

This choice can be understood in a simple way in the language of the traveling wave problem of Sec. 4. Note that in the ordered phase ($g > g_c$), starting from a low value (far from the cut off) $B_0 \ll g$, the recursion can at first be linearized. However, corresponding velocity will be negative, $V(g \gtrsim g_c) < 0$ as per Eq. (12). Hence, $B$ first increases exponentially as the depth $n$ increases until it reaches a stationary value due to the effect of the nonlinear cutoff $\sqrt{\epsilon^2 + B^2}$, see Eq. (5). Having $B_0 = \alpha_0 N^{-\phi}$ means that $B$ does not necessarily reach the nonlinear cutoff, even in the limit of large $n \approx \log_2 N$. Indeed, the linearized recursion gives, see Eq. (9):

$$\langle \ln B \rangle = -Vn + \ln B_0 - \frac{3}{2\gamma} \ln n + \langle \ln b_n \rangle$$

$$\approx -(V + \phi \ln 2)\,n - \frac{3}{2\gamma} \ln n + \text{Cste}\,, \tag{27}$$

with $\text{Cste} = \ln \alpha_0 + \langle \ln b \rangle$. This means that for $V + \phi \ln 2 > 0$, $\langle \ln B \rangle$ (with the log correction) decreases with $n$ and thus never reaches the upper cutoff value $g$ and a stationary regime. For our numerical observations, we chose $\alpha_0 = 100$ and $\phi = 1$. Thus for $g_c < g < g_e$, where $g_e = 2g_c \approx 0.27$ is given by $|V(g_e)| = \ln 2$, we can expect to find clear deviation from ergodic behavior.

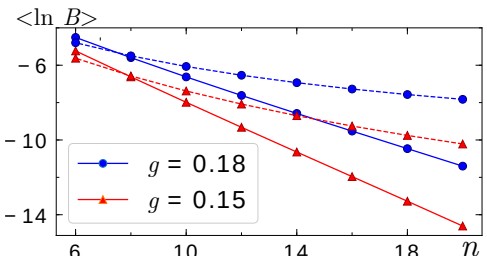

Figure 9: We show the variation of $\langle \ln B \rangle$ as a function of $n$, corresponding to different choices of the boundary field $B_0$ in the early ordered phase. The choice $B_0 = \frac{100}{N}$ (shown in solid lines) corresponds to a power law behavior indicating non-ergodicity. While, for the choice $B_0 = g$ (shown in dashed lines) we find an ergodic ordered phase, where $\langle \ln B \rangle$ tends to a stationary value at large $n$. Notice that for $n = 8$, we have $N = 766$ and $\frac{100}{N} = 0.13 \simeq g$, and so the dashed and plain curves of either color cross very close to that point.

In Fig. 9, we look at the contrasting behavior of $\langle \ln B \rangle$ as a function of $n$ for the two different choices of boundary field, $B_0 = g$ and $B_0 = 100/N$. We have looked at two different values of $g$ in the early ordered regime: $g = 0.15$ and $g = 0.18$. In the $B_0 = 100/N$ case, we observe a power law behavior of the typical field with the system size

$$\exp\langle \ln B \rangle \sim N^{D_1 - 1} \,. \tag{28}$$

By analogy with the algebraic behavior found for the typical value of $\mathfrak{J}G \sim N^{\mathcal{D}-1}$, with $0 < \mathcal{D} < 1$, in the non-ergodic delocalized phase of the Anderson localization transition on the Cayley tree [29,32,99], we interpret the behavior Eq. (28) as indicating a non-ergodic ordered phase. The exponent $0 < D_1 < 1$, analogous to the fractal dimension $\mathcal{D}$ for the Anderson localization problem, is introduced to characterize this property. For the system sizes $n \leq 20$, appropriately incorporating the log correction we find that the numerical data are well fitted by:

$$\langle \ln B \rangle + \frac{3}{2\gamma} \ln n \approx -[1 - D_1] n \ln 2 \,. \tag{29}$$

Recall that $\ln N \sim n \ln 2$. In the regime where linearization of the recursion is possible, $g < g_e$,

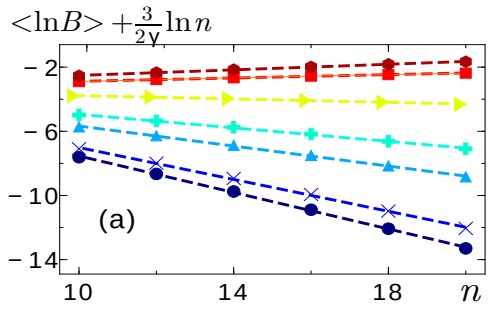
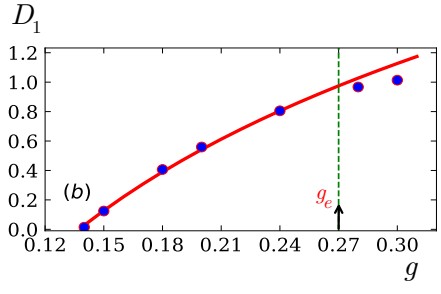

Figure 10: (a) $\langle \ln B \rangle + \frac{3}{2\gamma} \ln n$ (incorporating logarithmic corrections) as a function of $n$ is shown for selected values of $g = 0.3, 0.28, 0.24, 0.20, 0.18, 0.15$ and $0.14$ from top to below. The linear fits indicated by the dashed lines give the value of the corresponding exponent $D_1$ as per Eq. (29). (b) Exponent $D_1$ (as obtained from part (a) of the figure) as a function of $g$. We find a non-ergodic regime for $g < g_e \approx 0.27$, characterized by: $0 < D_1 < 1$. In the regime $g_c < g \lesssim g_e$ we find very good agreement with the theoretical prediction for $D_1$, Eq. (30) (red solid line).

we find from Eq. (27) with $\phi = 1$:

$$D_1 \approx -\frac{V}{\ln 2}. \tag{30}$$

In Fig. 10 we look at $\langle \ln B \rangle + \frac{3}{2\gamma} \ln n$ as a function of $n$ for different values of $g_c < g \leq 0.3$. We find a non-ergodic ordered regime characterised by: $0 < D_1 < 1$, where $D_1$ agrees very well with the theoretical value for $g_c < g < g_e$ and tends to unity as $g \to g_e$, similarly to what was observed in the non-ergodic delocalized phase of the Anderson transition in the Cayley tree.

A note is in order here, regarding the significance of finite-size effects in the present problem compared to the related Anderson problem. In the Anderson transition on random and tree graphs, drastic and non-trivial finite-size effects have been observed [32–34, 36, 37, 98, 102]. These effects were central to the debate concerning the existence of a non-ergodic delocalized phase [24–37, 40, 41, 54], which has now been resolved. In our problem, the mapping to the traveling wave problem allows us to analytically deduce the presence of a universal logarithmic finite-size correction that depends solely on the tail exponent $\gamma$, as indicated in Eq. (29). By incorporating this logarithmic correction, we can accurately determine the exponent $D_1$ which characterizes the non-ergodic ordered phase.

For the related problem of disordered hard-core bosons [94], a non-ergodic phase has also been identified, but for strong enough disorder, in the so-called Bose-glass regime. There, the *coherent fraction* (analogous to the ferromagnetic order parameter) is zero, while the one-body density matrix [103, 104] display some non-ergodic delocalized properties, with for instance an anomalously slow decay of the *condensed fraction*, suggesting a new form of off-diagonal quasi-long-range order induced by the disorder [94]. Here, since the $U(1)$ symmetry is not preserved by our quantum Ising Hamiltonian, we cannot further follow the analogy between the two models, namely there is no equivalent of the condensed fraction.

## 6 Strong spatial inhomogeneities

The different phases described above can show strongly inhomogeneous spatial distributions of cavity fields. Such spatial inhomogeneities are well known in the problem of directed polymers on the Cayley tree [66], which has a glassy phase at low temperature where the polymer explores only a finite number of branches instead of the exponentially many at disposal (the symmetry of replicas is then broken). In this last section, we characterize this property in the random field Ising model considered.

In Fig. 11, we contrast the spatial distribution of the cavity field within a given level $d$, in the disordered and ordered phases. We consider in each case a single realization of disorder and represent the corresponding spatial distribution of the cavity fields as follows. The position inside the finite Cayley tree with $n = 18$ is specified by the depth $0 < d < n-1$ (excluding the root and the boundary) increasing from the boundary to the root (where $d = n - 1$). Strong spatial inhomogeneity is clearly seen in the disordered case shown in Fig. 11(a) where the cavity fields amplitudes are large only on a few branches, sufficiently far from the homogeneous boundary. This is reminiscent to the behavior of the directed polymer problem. On the contrary, in the ordered case shown in Fig. 11(b), the spatial distribution is homogeneous, indicating an ergodic behavior. Finally, in the non-ergodic ordered case shown in Fig. 11(c), corresponding to a boundary condition $B_0 = 100/N$, the spatial distribution is again strongly inhomogeneous, similarly to the disordered phase.

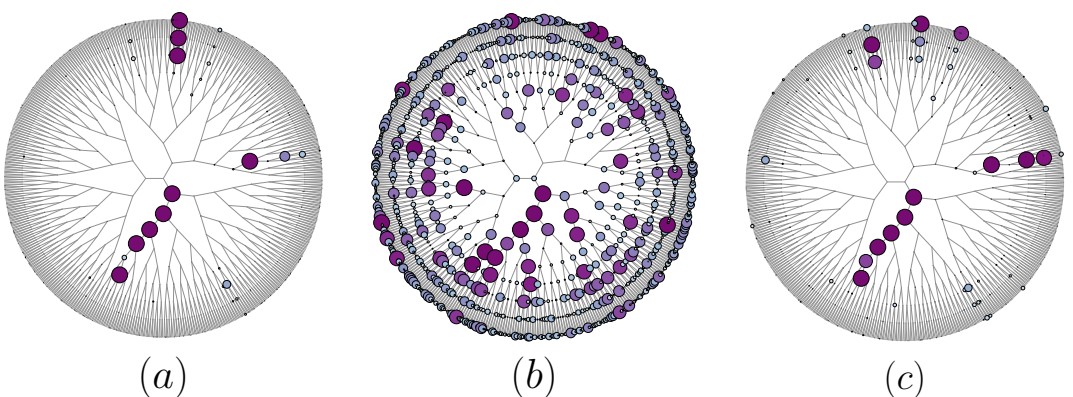

Figure 11: Spatial distribution of the cavity fields on the vertices of a Cayley tree far away from the uniform boundary, for a single random disorder configuration. The total number of generations is $n = 18$. Only generations $9 \leq d \leq 18$ are shown (from the root to half of the tree). The radius of the circles at vertex $i$ of generation $d$ are given by: $[B_d^{(i)} - \min_i(B_d^{(i)})]/[\max_i(B_d^{(i)}) - \min_i(B_d^{(i)})]$, where the min and max are taken over all the sites within a given level $d$. Panel (a) corresponds to $g = 0.03$, deep in the disordered phase, while panel (b) corresponds to $g = 0.3$, deep in the ordered phase. Both have boundary condition $B_0 = g$. In panel (c) $g = 0.18$ and $B_0 = 100/N$. It describes the non-ergodic delocalized phase. Note that the same randomness was used in these three simulations. In (a) and (c), cavity fields have large amplitudes only on few branches, while (b) shows an ergodic delocalized behavior.

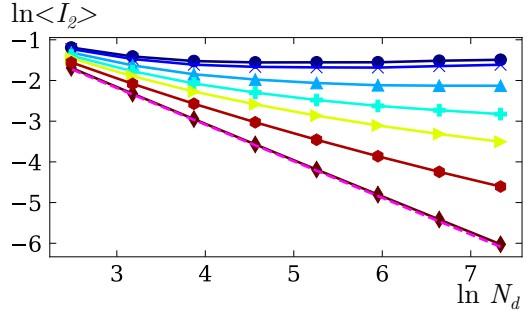

Figure 12: The second moment $\langle I_2(d = n/2) \rangle$ of the cavity fields $\tilde{B}_d^i$ at depth $d = n/2$, as a function of the total number of site $N_{n/2}$ at depth $d = n/2$, see Eq. (31), shown in log scale for $B_0 = 100/N$. The values of $g$ considered are: $0.14, 0.15, 0.18, 0.2, 0.24, 0.3$ and $0.45$, from the top to bottom. In the non-ergodic regime ($g_c < g < g_e$), the second moment tends to a constant, showing that cavity fields explore only a few branches of the tree. The ergodic behavior $\langle I_2(n/2) \rangle \sim 1/N_{n/2}$ is restored for $g > g_e \approx 0.27$. This is indicated by the dashed line shown for the lower curve corresponding to $g = 0.45$.

To characterize these non-ergodic features more quantitatively, we study the normalized second moment of the measure given by the cavity fields within a level $d$:

$$I_2(d) = \frac{\sum_l \left(B_d^{(l)}\right)^2}{\left(\sum_l B_d^{(l)}\right)^2}, \tag{31}$$

where the sum is over all the sites at a given level $d$. $I_2(d)$ is analogous to the inverse participation ratio and gives the inverse of the number of sites occupied by the cavity fields. If one

calls $N_d$ the number of sites at a level $d$, then one can see that $I_2(d) = 1/N_d$ if all the cavity fields are equal, and $I_2(d) = 1$ if one cavity field is much bigger than all the others. Then, the study of $\langle I_2(d) \rangle$ as a function of $N_d$, gives a characterization of non-ergodic features in the model we study. We choose to study $\langle I_2(n/2) \rangle$ as a function of $N_{n/2}$, at the level $d = n/2$, deep inside the tree, far enough from the boundary, when the boundary condition is $B_0 = 100/N$ and for $g > g_c$.

Fig. 12 shows that the second moment tends to a constant in the non-ergodic regime, corresponding to small $g_c < g < g_e$ values. As the second moment corresponds to the inverse of the number of sites with large cavity fields, this behavior indicates that the cavity fields explore only a few branches of the tree in the non-ergodic regime, confirming our previous observations, see Fig. 11(c), and the analogy between this phase and the directed polymer problem [66], see Eq. (27). At large $g \gtrsim g_e \approx 0.27$ values, on the other hand, the second moment has a $N_d^{-1}$ dependence characteristic of an ergodic regime.

# 7 Conclusion

In this paper we have investigated the zero temperature quantum paramagnetic-ferromagnetic phase transition for a random transverse field Ising model in its cavity mean field description by Feigel'man, Ioffe and Mézard [70] on the Cayley tree. This problem was initially considered to describe the mechanism responsible for the strong spatial inhomogeneities observed in strongly disordered superconductors [70, 89, 90]. In the Cayley tree, the disordered phase of this system is analogous to the problem of directed polymers and therefore has a glassy regime characterized by strong inhomogeneities. This is not unlike the localized phase of the Anderson transition, which has similar properties [23, 25, 33, 45]. The emergence of such non-ergodic properties due to disorder has been the subject of great interest recently, in particular with regard to the existence of a non-ergodic delocalized phase [24–37, 40, 41, 54] and in connection with the problem of many-body localization [4, 6, 10–12, 46, 48–53].

We addressed two open questions in this context: We first described the critical properties of the paramagnetic-ferromagnetic transition for the random transverse-field Ising model, and examined the possibility of a non-ergodic ordered phase. One of the motivations for the first question is the similarity between this problem and the Anderson transition on the Cayley tree whose critical properties are not yet well understood [29, 33, 36, 38, 41]. The cavity mean-field equation can be seen as a simpler form of the self-consistent equation for Anderson localization [20], thus allowing a better understanding of these critical properties. Similarly, we use the analogy with the Anderson transition, which has a non-ergodic delocalized phase on the finite Cayley tree [24–29, 32], to understand the conditions for the existence of a non-ergodic phase. Such a phase has recently been observed in disordered hard core bosons on the Cayley tree [94]. The system we have considered can be seen as an approximation of this problem [70, 89, 91].

To answer these questions, we have followed an analogy with a traveling wave problem corresponding to a branching random walk in presence of an absorbing wall [69, 97] and direct numerical simulations of the cavity mean field equation on a finite Cayley tree. As in the case of the Anderson transition on the finite Cayley tree, we find that the nature of the ordered phase (corresponding to the delocalized phase) depends crucially on the boundary conditions.

We have first considered constant boundary conditions, which corresponds to a transition between a disordered and an ergodic ordered phase. We have tested with great precision the results deriving from the expected analogy with the problem of traveling wave with a wall [69, 97]. We have found quantitative agreement with the predictions in the paramagnetic

phase which can be identified with the traveling wave regime by appropriately incorporating the logarithmic corrections for the small system sizes considered. We therefore show that such corrections can be very important for numerical studies with finite size systems. The critical behavior and in particular the cumulative distribution function agrees very well with the mathematical predictions considered in [69,97], in particular for the critical exponent $\delta = 1/3$. The critical properties have been assessed using a detailed finite-size scaling approach incorporating irrelevant corrections. We have confirmed the critical exponent predicted in [97], $\nu \approx 3/2$.

Furthermore, we have explored the appropriate choice of boundary condition for observing the much studied non-ergodic delocalized phase [24–37, 40, 41, 54] from a traveling wave perspective. To be able to observe a non-ergodic ordered phase, it is necessary to take boundary conditions which vanish algebraically with the volume of the system, $B_0 \sim N^{-\phi}$ where $\phi$ is a positive exponent. In this case, the ordered phase is characterized by an order parameter which also vanishes algebraically, characterized by a fractal dimension which reaches one in the ergodic phase. We have characterized the strong spatial inhomogeneities of this phase.

Our work has revealed an intriguing analogy with the Anderson transition, establishing connections through the traveling/non-traveling phase transition [97]. However, it is important to acknowledge the limitations of this analogy. Specifically, our focus is on ground state properties, whereas the Anderson problem primarily addresses highly excited states. An extension of the CMF approach we have used is required to describe properties related to excitations above the ground state, including the presence of a mobility edge or other dynamical properties [70].

An interesting outcome of the analogy between CMF and the Anderson problem lies in the analytical predictions it provides for critical properties. Notably, our results put forth analytical predictions for critical behavior and finite-size scaling, which, to the best of our knowledge, have not been explored within the framework of the Anderson problem. These predictions could make a substantial contribution to resolving debates surrounding critical exponents of the Anderson transition on random graphs [29, 29, 32, 33, 36, 41, 98].

Furthermore, this analogy allows us to explore the extension of non-ergodic properties to systems with interactions, shedding new light on the non-ergodic behaviors observed in closely related models like hard-core disordered bosons on the Cayley tree [94]. An open question remains regarding whether, similar to the Anderson transition, the presence of a non-ergodic delocalized phase crucially depends on the type of graphs considered [28, 32, 33, 105], such as Random Regular Graphs or small-world networks instead of the Cayley tree. Our work reveals a distinction in nature between the ordered phase of the infinite Bethe lattice/Cayley tree, which is ergodic, and a finite Cayley tree where the delocalized phase is non-ergodic. It essentially boils down to a matter of the order of limits. Due to the finite boundary fraction compared to the total volume on a finite Cayley tree, the order of limits $B_0 \to 0$ and $N \to \infty$ becomes crucial. In the usual order, where $N \to \infty$ precedes $B_0 \to 0$ (corresponding to the infinite Bethe lattice), the ordered phase exhibits ergodic behavior. Conversely, in the inverted order, where $B_0 \to 0$ precedes $N \to \infty$ (corresponding to $B_0 \sim N^{-\phi}$), the non-ergodic ordered phase emerges.

Since the approach we have used is based on a non-trivial mean-field approximation, it is important to compare its outcomes with other approaches. Strong-disorder renormalization procedures have been used in such graphs of effective infinite dimensionality [82–84, 106]. References [82,84,106] point towards a transition of infinite randomness criticality, but where the randomness grows only logarithmically with $N$. It remains unclear to us whether the CMF approach aligns with this logarithmic form of infinite randomness. On the one hand, the CMF approach predicts that, at the critical point, both the typical and averaged values of $B$ exhibit stretch exponential decay. While the typical decay mirrors the behavior observed in the conventional 1D infinite randomness criticality, the averaged decay contradicts the expected

power-law behavior in such cases. On the other hand, Griffiths effects have been observed in the CMF approach on both sides of the transition [70]. The strong disorder renormalization approach of Ref. [83] establishes a connection with the directed polymer problem in the paramagnetic phase, thus confirming the analogy with the traveling phase we have described. Lastly, CMF indicates the existence of an infinite disorder fixed point in dimensions $d = 2$ and 3 [88]. These contrasting findings underscore the need for further investigations into the nature of infinite randomness criticality in systems of effective infinite dimensionality.

Other interesting perspectives include expanding our analysis to describe other quantum many-body systems on the Cayley tree, for example the metal-insulator transition for disordered fermions in the statistical dynamical mean-field approach [107, 108]. This type of system can now be studied experimentally, for example by means of quantum computers [109]. The study of their non-ergodic properties should make it possible to understand the emergence of similar properties in finite dimensionality.

## Acknowledgments

AC and GL wish to thank J. Gong and C. Miniatura for fruitful discussions. We also thank José Hoyos for interesting discussions.

**Funding information** This study has been supported through the EUR grant NanoX n° ANR-17-EURE-0009 in the framework of the "Programme des Investissements d'Avenir", research funding Grants No. ANR-17-CE30-0024, ANR-18-CE30-0017 and ANR-19-CE30-0013, and by the Singapore Ministry of Education Academic Research Fund Tier I (WBS No. R-144-000-437-114). We thank Calcul en Midi-Pyrénées (CALMIP) for computational resources and assistance.

## A  Analytical description of the traveling–non-traveling phase transition

In order for the paper to be sufficiently self-contained, we recall in this appendix the analytical derivations of the main results on which we relied in Sec. 4.

This appendix is about travelling waves in the universality class of the Fisher-KPP equation [67, 68], invading an unstable phase on the right from a stable phase on the left. As explained below, there are several possible travelling waves, with different velocities $v$ and spatial decay rates $\lambda$, but the most relevant one for our problem is the *critical travelling wave* going at velocity $v_c$ and having a spatial decay rate $\lambda_c$.

These critical values $v_c$ and $\lambda_c$ appear directly in some important results of the paper, but under another name:

$$V(g) \equiv -v_c, \qquad \gamma \equiv \lambda_c. \tag{A.1}$$

See in particular Eq. (9), Eq. (11), etc.

### A.1  A Fisher-KPP equation

We discuss first how the linearized cavity mean field recursion Eq. (7) describing the disordered phase can be mapped onto the traveling wave problem described in [66]. In this analogy, the distance $d$ from the boundary plays the role of a time.

Assume that $B_d$ obeys Eq. (7) with $B_0$ being some non-random constant. Notice first that $B_d/B_0$ is independent of $B_0$ and introduce

$$G(x,d) := \left\langle \exp\left[ -e^{-x} B_d/B_0 \right] \right\rangle, \tag{A.2}$$

where $\langle \cdot \rangle$ represents averaging over the random variable $B_d$. Using Eq. (7), we obtain the evolution equation

$$G(x, d+1) = \left\langle G\left( x - \ln\frac{g}{K} + \ln|\epsilon|, d \right) \right\rangle^K, \tag{A.3}$$

where $\langle \cdot \rangle$ represents now averaging over the random variable $\epsilon$.

There are two constant solutions to Eq. (A.3): $G(x,d) = 0$ and $G(x,d) = 1$. We say that $G = 0$ is the *stable* phase and $G = 1$ is the *unstable* phase; indeed, if $G(x,0)$ is a constant in $(0,1)$, then $G(x,d)$ is at all time $d$ a constant, converging to 0 as $d \to \infty$.

Eq. (A.3) describes the propagation of a front between the stable phase $G(-\infty, d) = 0$ and the unstable phase $G(\infty, d) = 1$. This equation is in the universality class of the Fisher-KPP equation [67, 68], first introduced in 1937:

$$\frac{\partial F}{\partial t} = \frac{\partial^2 F}{\partial x^2} - F(1-F). \tag{A.4}$$

(Nota: the Fisher-KPP equation is often written for $f = 1 - F$; the quantity $f$ follows the same equation as $F$, except that the minus sign in front of $F(1-F)$ is changed into a plus sign.)

As for Eq. (A.3), Eq. (A.4) has two uniform solutions, $F = 0$ which is stable and $F = 1$ which is unstable. For an initial condition that interpolates between 0 and 1, a front builds up where the stable phase 0 invades the unstable phase 1. While not obvious at first sight, Eq. (A.3) and Eq. (A.4) share enough features to behave in similar ways: the expectation over $\epsilon$ in Eq. (A.3) plays the role of the diffusion in Eq. (A.4). The reaction $-F(1-F)$ term in Eq. (A.4) leaves $F = 0$ and $F = 1$ unchanged, and pushes any intermediate values towards 0; similarly, taking the $K$-th power in Eq. (A.3) leaves $G = 0$ and $G = 1$ unchanged, but pushes intermediate values towards 0.

There is an exhaustive literature on the Fisher-KPP and related equations; in particular, much work has been devoted on giving precise asymptotics on the position of the front [110–115]. More references can be found in a review from 2003 [116] and in a dissertation from 2016 [117].

In the following sections, we review quickly and apply to Eq. (A.3) some well-known results of Fisher-KPP related equations and obtain in that way some informations on $B_d$.

## A.2 Properties of travelling wave solutions

We first consider *travelling wave solutions with velocity $v$* to Eq. (A.3). These solutions satisfy, for some function $\omega_v$,

$$G_{\text{TW}}(x,d) = \omega_v(x - vd). \tag{A.5}$$

Since, from Eq. (A.2), the quantity $G(x,d)$ for fixed $d$ is an increasing function of $x$ going from $G(-\infty, d) = 0$ to $G(+\infty, d) = 1$, we only consider travelling wave solutions satisfying $\omega_v \in (0,1)$, $\omega_v(-\infty) = 0$ and $\omega_v(\infty) = 1$.

Such a travelling solution exists for all $v \geq v_c$, where $v_c$ is called the *critical velocity*. Notice that the problem is translation invariant; this implies that if $\omega_v(x)$ represents a travelling wave solution, then $\omega_v(x + a)$ for any constant $a$ is also a travelling wave solution.

Plugging Eq. (A.5) into Eq. (A.3), one obtains that the travelling waves $\omega_v$ satisfy

$$\omega_v(x - v) = \left\langle \omega_v(x - \ln\frac{g}{K} + \ln|\epsilon|) \right\rangle^K. \tag{A.6}$$

We linearize for large $x$, where $\omega_v$ is close to the unstable solution 1. Looking for exponential solutions

$$1 - \omega_v(x) \propto e^{-\lambda x}, \quad \text{as } x \to \infty, \tag{A.7}$$

one obtains a relation between the velocity $v$ and the spatial decay rate $\lambda$:

$$v = v(\lambda) := \ln \frac{g}{K} + \frac{1}{\lambda} \ln \left[ K \left\langle e^{-\lambda \ln |\epsilon|} \right\rangle \right]. \tag{A.8}$$

The critical velocity $v_c$ is the minimal value reached by $v(\lambda)$:

$$v_c = v(\lambda_c) = \min_\lambda v(\lambda). \tag{A.9}$$

The travelling waves $\omega_v(y)$ for $v > v_c$ behave as in Eq. (A.7), where $\lambda$ satisfies Eq. (A.8) for the given value of $v$ and $\lambda < \lambda_c$. However, the critical travelling wave $\omega_c$ (for $v = v_c$) has a polynomial prefactor:

$$1 - \omega_c(x) \simeq C(x + a)e^{-\lambda_c(x+a)}, \quad \text{as } x \to \infty, \tag{A.10}$$

for some constant $C$ which is hard to compute (it depends on the whole non-linear equation Eq. (A.6)). The value of $a$ is arbitrary; it depends on the choice of the travelling wave, which is only defined up to translation.

In this paper, we mostly consider the case $K = 2$ and $\epsilon$ uniform in $[-1, 1]$. Then Eq. (A.8) becomes, for $\lambda \in (0, 1)$:

$$v(\lambda) = \ln \frac{g}{2} + \frac{1}{\lambda} \ln \frac{2}{1 - \lambda}. \tag{A.11}$$

Clearly, $\lambda_c$ does not depend on $g$. Numerically, one finds that

$$\lambda_c \simeq 0.626635. \tag{A.12}$$

The critical velocity depends on $g$. Numerically,

$$v_c \simeq \ln \frac{g}{2} + 2.67835 = \ln \frac{g}{g_c}, \quad \text{with} \quad g_c \simeq 0.137353. \tag{A.13}$$

## A.3 Properties of the front

We only consider the case $K = 2$ and $\epsilon$ uniform in $[-1, 1]$. Going back to Eq. (A.2), one gets for $d = 0$

$$G(x, 0) = \exp\left[-e^{-x}\right] \simeq 1 - e^{-x}, \quad \text{as } x \to \infty.$$

Thus, the initial condition converges to 1 as $x \to \infty$ much faster than does the critical travelling wave, see Eq. (A.10) and Eq. (A.12). In this case, it is known that the front properly centered converges uniformly to the critical travelling wave: there exists a choice of $\omega_c$ such that

$$G\left(v_c d - \frac{3}{2\lambda_c} \ln d + x, d\right) \to \omega_c(x), \quad \text{as } d \to \infty. \tag{A.14}$$

(Recall that any translation of a travelling wave is another travelling wave. The equation above selects one of these travelling waves.)

Note that for other choices of $K$ and $\epsilon$, we could find $\lambda_c > 1$. Then, the initial condition would converge to 1 more slowly than the critical travelling wave. In that case, the behaviour of the front would be rather different. In the following, we assume to always be in the case $\lambda_c < 1$.

### A.4 Prediction for the disordered phase

From the definition Eq. (A.2) of $G$, one has

$$G\left(v_c d - \frac{3}{2\lambda_c}\ln d + x, d\right) = \left\langle \exp\left[-e^{-x}b_d\right]\right\rangle, \tag{A.15}$$

where $b_d$ is defined by

$$B_d = B_0 \exp\left[v_c d - \frac{3}{2\lambda_c}\ln d\right]b_d. \tag{A.16}$$

Then, Eq. (A.14) means that the distribution of $b_d$ satisfies

$$\left\langle \exp\left[-e^{-x}b_d\right]\right\rangle \to \omega_c(x), \quad \text{as } d \to \infty. \tag{A.17}$$

This means $b_d$ converges (in distribution) to a random variable $b = b_\infty$ with generating function given by

$$\left\langle \exp\left[-sb\right]\right\rangle = \omega_c(-\ln s). \tag{A.18}$$

When $s > 0$ is small, Eq. (A.10) gives that

$$\left\langle \exp\left[-sb\right]\right\rangle \simeq 1 - Ce^{-\lambda_c a}(-\ln s + a)s^{\lambda_c}. \tag{A.19}$$

(The value of $a$ is no longer arbitrary as Eq. (A.17) only holds for one specific choice of $\omega_c$, but determining its value is hard.) By usual tail analysis, this implies that for large $b$ one has

$$\text{Prob}(b > x) \simeq (C'\ln x + C'')x^{-\lambda_c}, \tag{A.20}$$

where $C'$ and $C''$ are other constants.

To summarize, Eq. (A.16) gives the typical value of $B_d$ for large $d$ in the linearized cavity mean field recursion Eq. (7), while Eq. (A.20) gives the tail distribution of large values of $B$.

Note that, to obtain Eq. (7), we first assumed that $B_d \to 0$ as $d \to \infty$. This is consistent with Eq. (A.16) only if $v_c < 0$ which is the case if $g < g_c$, see Eq. (A.13).

With the substitutions $V(g) \equiv -v_c$ and $\gamma \equiv \lambda_c$, then Eq. (A.12) and Eq. (A.13) are equivalent to Eq. (12) and Eq. (13). Furthermore, Eq. (9) is the same as Eq. (A.16) in the large $d = n - 1$ limit where $b_d \simeq b$, and Eq. (11) is obtained by differentiating Eq. (A.20).

### A.5 Slow travelling waves

We mentioned that travelling waves satisfying $\omega_v \in (0, 1)$, $\omega_v(-\infty) = 0$ and $\omega_v(\infty) = 1$ only exist for $v \geq v_c$. It turns out that there exists travelling waves for $v < v_c$ if one allows $\omega_v$ to take values larger than 1. As those travelling waves are relevant to the study of the disordered/ordered transition, we now quickly review their properties.

From now on, we use the following notation:

$$x \lll y \text{ or } y \ggg x, \quad \text{to mean that} \quad y - x \quad \text{is large.} \tag{A.21}$$

From Eq. (A.9), one has that $v(\lambda_c) = v_c$, $v'(\lambda_c) = 0$ and $v''(\lambda_c) \geq 0$. Actually, in the case $K = 2$ and $\epsilon$ uniform in $[-1, 1]$, one obtains from Eq. (A.11)

$$v''(\lambda_c) \simeq 11.4477. \tag{A.22}$$

For $\lambda$ close to $\lambda_c$, one has therefore

$$v(\lambda) \simeq v_c + \frac{v''(\lambda_c)}{2}(\lambda - \lambda_c)^2. \tag{A.23}$$

One can obtain $v(\lambda) < v_c$ by giving a small imaginary part to $\lambda$. Let $L$ be some large parameter, and take

$$\lambda = \lambda_c \pm \frac{i\pi}{L}. \tag{A.24}$$

Then, in (A.23),

$$v(\lambda) \simeq v_c - \frac{v''(\lambda_c)\pi^2}{2L^2}. \tag{A.25}$$

It turns out that travelling waves $\omega_v$ with $v < v_c$ given by Eq. (A.25) exist, and that their large $x$ behavior is obtained by using Eq. (A.24) in Eq. (A.7). One obtains

$$1 - \omega_v(x) \simeq \frac{CL}{\pi} \sin \frac{\pi(x+a)}{L} e^{-\lambda_c(x+a)}, \quad \text{if } x + a \gg 0, \tag{A.26}$$

where $C$ has some definite value, but $a$ is arbitrary. (Recall that a shifted travelling wave is another travelling wave.) Notice that this front oscillates around the unstable phase 1. Notice also that by sending $L$ to infinity, one recovers Eq. (A.10); in particular the $C$ in Eq. (A.26) is the same as the $C$ in Eq. (A.10).

## A.6 Prediction for the ordered phase

Assume now that $g > g_c$, with $g$ very close to $g_c$. Eq. (A.13) gives $v_c > 0$ and Eq. (A.16) would predict that $B \to \infty$, but of course this is not true as Eq. (5) implies that $B_d < g$. We see that $B_d$ must reach some stationary distribution $B$ as $d \to \infty$.

We introduce the function $G$ as in Eq. (A.2) for the long time distribution $B$, but without the $B_0$ term which no longer makes sense and, of course, without the dependence in $d$:

$$G(x) := \left\langle \exp\left[-e^{-x}B\right] \right\rangle. \tag{A.27}$$

When $x$ is a large positive number, given that $B$ is bounded, one can expand the exponential to obtain

$$G(x) \simeq 1 - \langle B \rangle e^{-x}, \qquad \text{for } x \gg 0. \tag{A.28}$$

When $x$ is a large negative number, the value of $G$ is dominated by the small values of $B$ (of order $e^x$). But for these small values, the linearized recursion Eq. (7) still holds, and this implies that the recursion Eq. (A.3) (without the $d$) still holds:

$$G(x) \simeq \left\langle G\left(x - \ln\frac{g}{K} + \ln|\epsilon|\right) \right\rangle^K, \quad \text{for } x \ll 0, \tag{A.29}$$

but Eq. (A.29) is actually the same as Eq. (A.6), the equation defining the travelling wave $\omega_v$, for $v = 0$. We conclude that

$$G(x) \simeq \omega_0(x), \quad \text{for } x \ll 0, \tag{A.30}$$

for some choice of the zero-velocity travelling wave $\omega_0$. (Recall that travelling waves are defined up to some translation.)

Recall from Eq. (A.13) that $v_c = \ln(g/g_c)$. When $g \simeq g_c$, this gives $v_c \simeq (g - g_c)/g_c$. Then, from Eq. (A.25), the choice of $L$ which makes $v = 0$ is

$$L \simeq \sqrt{\frac{v''(\lambda_c)\pi^2 g_c}{2}} \, (g - g_c)^{-1/2}. \tag{A.31}$$

This $L$ gives the length of the sine arches in the travelling wave $\omega_0$, see Eq. (A.26). To summarize, we have at this point the following prediction for $G(x)$:

$$G(x) \simeq \begin{cases} 1 - \langle B \rangle e^{-x}, & \text{for } x \gg 0, \\ 1 - \frac{CL}{\pi} \sin \frac{\pi(x+a)}{L} e^{-\lambda_c(x+a)}, & \text{for } -a \ll x \ll 0, \\ 0, & \text{for } x \ll -a. \end{cases} \tag{A.32}$$

(More precisely, $G(x)$ interpolates from 0 to 1 in a region of size of order 1 around $x = -a$.)

Notice that, at this point, we still have to determine the correct value of $a$. (Recall that $a$, which comes from Eq. (A.26), represents the translation symmetry.) This value of $a$ will be determined in a self-consistent way later on.

We claim that Eq. (A.32) implies that the distribution of $\ln B$ is concentrated around $-a$, with a tail given by

$$\text{Prob}(\ln B > u - a) \simeq \hat{C} L \sin \frac{\pi u}{L} e^{-\lambda_c u}, \quad \text{for } 0 \ll u \ll a, \tag{A.33}$$

for some constant $\hat{C}$. To do this, we check that this form gives the correct function $G(x)$ in Eq. (A.32). Write

$$
\begin{aligned}
G(x) &= \int_0^\infty dy\, (-)\frac{d}{dy}\Big[\text{Prob}(B > y)\Big] e^{-e^{-x}y} \\
&= 1 - \int_0^\infty dy\, \text{Prob}(B > y) e^{-x - e^{-x}y} \\
&= 1 - \int du\, e^u \text{Prob}(B > e^u) e^{-x - e^{u-x}}.
\end{aligned}
\tag{A.34}
$$

But $\text{Prob}(B > e^u) = \text{Prob}(\ln B > u)$. Changing $u$ into $u - a$ and $x$ into $x - a$, we get

$$G(x - a) = 1 - \int du\, \text{Prob}(\ln B > u - a) e^{u - x - e^{u-x}}. \tag{A.35}$$

The function $u \mapsto \text{Prob}(\ln B > u - a)$ is 1 for $u \ll 0$ and roughly $e^{-\lambda_c u}$ for $0 \ll u \ll a$, and 0 for $u > a$. One can then check that the integral in Eq. (A.35) is dominated by $u$ close to $x$ if $x \ll a$. If $x \ll 0$, then the term $\text{Prob}(\ln B > u - a)$ can be replaced by 1 in the integral. If $0 \ll x \ll a$, it can be replaced by its expression Eq. (A.33) with $\sin(\pi u/L)$ replaced by $\sin(\pi x/L)$, as the difference between $x$ and relevant values of $u$ is small compared to $L$. Then, one checks that these substitutions lead respectively to the third and second line of Eq. (A.32). We already argued that the first line must hold (independently of the distribution of $B$, as long as it is bounded) in Eq. (A.28). This validates the claim that $\text{Prob}(\ln B > u - a)$ is given by Eq. (A.33).

We still have to determine $a$. Recall that $B$ is bounded from above by $g$; this means that $\text{Prob}(\ln B > u - a)$ should cancel very quickly when $u$ becomes close to $a$. But the form Eq. (A.33) cancels when $u = L$; this suggests that $a$ should be close to $L$.

To summarize, in the regime $g > g_c$ with $g$ close to $g_c$, we expect the stationary distribution of $\ln B$ to be concentrated around $-L$, with $L \gg 1$ given by (A.31). This distribution of $\ln B$ has a power-law tail with an exponent $-\lambda_c$ modulated by a sine arch, as given in Eq. (A.33) with $a = L$. Fig. 13 illustrates this prediction:

## A.7 At criticality

We now consider $g = g_c$. The results for $g < g_c$ and $g > g_c$, as exposed in this appendix, are reminiscent of the problem studied [97] of a Fisher front $F(u, t)$ with a drift and a boundary condition $F(0, t) = 0$ (with 0 being now the unstable phase). The results of [97] for the front they study are exactly the same as the properties of $\text{Prob}(\ln B_d > u)$ (with $d$ playing the role of time) that we have just presented: depending on the drift (which plays the same role as $g$), the front in [97] escapes to infinity (as in our disordered phase), or reaches a stationary state with a sine arch (as in our ordered phase).

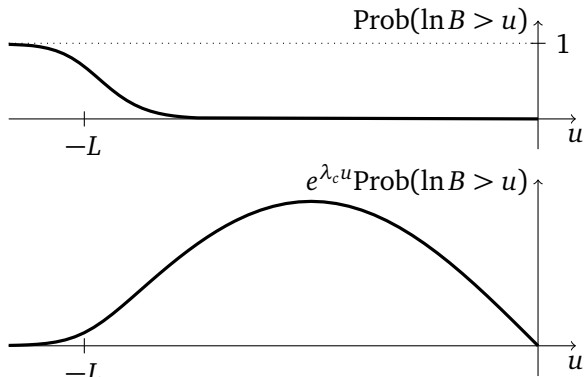

Figure 13: Sketch of the distribution of $\ln B$ in the stationary regime of the ordered phase.

This remark suggests that $\text{Prob}(\ln B_d > u)$ has in fact all the properties of a Fisher front with a boundary condition. Then, if the analogy still holds at criticality, [97] gives a prediction for the problem we consider, which we now present.

For $g = g_c$, the front escapes at infinity, meaning that $B_d$ goes to zero as $d \to \infty$. More precisely, for large $d$, The distribution of $\ln B_d$ is concentrated around $-L_d$, with

$$L_d \simeq \left( \frac{3}{2} \pi^2 v''(\lambda_c) d \right)^{1/3}, \tag{A.36}$$

and that, similarly to (A.33),

$$\text{Prob}(\ln B_d > u - L_d) \simeq \hat{C} L_d \sin \frac{\pi u}{L_d} e^{-\lambda_c u}, \tag{A.37}$$

for $0 \lll u \lll L_d$ and $d$ large.

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
