# Peer review of "Traveling/non-traveling phase transition and non-ergodic properties in the random transverse-field Ising model on the Cayley tree"

_SciPost Physics, doi:SciPost Phys. 15, 211 (2023)_

## Round 1 · Referee Report · Anonymous (Referee 1) · 2023-4-19

Strengths

All the results are very interesting and explained in detail with many figures,
while the Appendix is very useful to make the traveling-wave analysis self-contained.

Weaknesses

None

Report

I strongly recommend the publication of this preprint in the present form.
Congratulations to the authors for their great work !

Requested changes

No requested changes

  • validity: top
  • significance: top
  • originality: top
  • clarity: top
  • formatting: perfect
  • grammar: excellent

Author:  Ankita Chakrabarti  on 2023-09-22  [id 3999]

(in reply to Report 1 on 2023-04-19)

Our response to this Report is contained in the attached file.

Attachment:

Reply.pdf

---

## Round 1 · Referee Report · Ivan Khaymovich (Referee 2) · 2023-5-16

Strengths

1 - Interesting topic
2 - Combination of matching analytical and numerical results

Weaknesses

1 - Hard to follow the logic of the text,
2 - One have to address Appendix during reading,
3 - Possible mapping to free-fermion models?
4 - Unclear finite-size effects

Report

The authors address a random transverse field Ising model on a Bethe lattice and investigate the Anderson and ergodicity breaking phase transitions, using travelling/non-travelling phase transition as an analogue.

Concerning the manuscript I have several questions and comments:

1) On of the main concerns is related to the fact that Jordan-Wigner transformation for the random transverse field Ising model in 1d is known to be mapped to the free-fermion system with some induced superconductivity (a quadratic Hamiltonian, conserving only particle-number parity). From the first glance, it seems that the same construction should be valid also for the Bethe lattice, as there are no loops on such graph and, thus, one can easily sort the sites in a certain 1d ordering. As soon as one can map the system to the non-interacting fermionic one, it seems natural to expect that the model will be in the same universality class as the Anderson model on the Bethe lattice. Please comment on whether it is the case and if so, what you predict about the critical exponents for the corresponding Anderson model on the Bethe lattice.

2) If the authors still claim that their model is interacting and cannot be mapped to any non-interacting model, then there appears another question, related to the boundary conditions (bare regularizer $B_0$). A mentioned in [30] for the Rosenzweig-Porter model, by scaling $B_0 \sim N^{-\phi}$, with the number of sites on the graph, one can address not only Anderson localization transition, but also the ergodicity-breaking one. Why does the same scaling procedure work for the interacting model in focus?

3) As a follow-up comment, I would like to emphasize that in [30] the scaling of the regularizer with $N$ was important in order to find the scaling of the mini-band width (Thouless energy), which in the non-ergodic extended phase scaled down with a certain $\phi>0$. At the same time it is known that in the Anderson model on Bethe lattice and/or on the random regular graph, the Thouless energy does not scale with $N$ and stays finite (but small with respect to the bandwidth) up to the Anderson transition. Please comment on why do you need to take $B_0\sim N^{-\phi}$? Do you expect to have the corresponding Thouless energy scaling down with $N$?

4) If the authors claim that the $B_0\sim N^{-\phi}$ in the non-ergodic extended phase is related to the $N$-scaling of the Thouless energy, they should consider either the overlap correlation function $K(\omega)$ or the local density of states, showing the corresponding miniband structure, like in the Rosenzweig-Porter.

5) The random regular graph is known to have drastic finite-size effects, while in the current model (which seemed to be mapped to a very similar model) the authors seem to overcome this issue. Please comment on the finite-size effects in the model in focus and on the possible influence of them on the (numerical) results. Especially this question should be asked to the claimed presence of the non-ergodic extended phase. In order to clarify this, please show the drift of the extrapolated fractal dimension $D_1$ in Fig. 10, fitted from a sliding window over system sizes.

6) The Anderson model on the random regular graph is known to show the mobility edge behavior. Please comment on the presence of the mobility edge in the model in focus and, in the case of its presence, please clarify which averaging over the eigenenergies has been taken to calculate fractal dimensions and other measures of the transitions. It might happen that the energy-resolved measures are needed for the case in focus. This is especially important, taking into account spatial inhomogeneity of the model, discussed in Sec. 7, as it can imply some spectral inhomogeneities as well.

7) In Figure 3 deeply in the ordered (delocalized) phase, there is an apparent bimodal distribution $P(\ln B)$. Please comment on the origin of this bi-modality.

8) In addition to the previous questions and comments, I would like to draw the authors' attention to different values of the branching number $K$, especially to the small-world networks, considered for the case of the Anderson model on the random regular graph by some of the authors of this manuscript. What do you expect to see for $1<K<2$ in the considered model? What are the peculiarities of this model?

To sum up, I find the topic of the manuscript rather interesting and valuable for the community, but the current version of the manuscript should be significantly improved in order to reach high standards of SciPost Physics.

Please answer all my questions and address all comments and suggestions. After that, if I am asked, I will provide my final opinion on the manuscript.

Requested changes

Main requested changes: please see the report above.

Minor changes: a - Please clarify in the abstract what "constant and algebraically vanishing boundary conditions" mean: it is unclear, while reading for the first time.

b-References [30] and [55] are mostly devoted to the Rosenzweig-Porter model, but not to the Bethe lattice or random regular graph: I am not sure that [30] is correctly cited in several places, as well as [55].

c - The reference list on a non-ergodic delocalized phase are far from being complete: the works on Gaussian Rosenzweig-Porter model contain not only [30] and [55], but also - mathematical proof of it https://doi.org/10.1007/s11005-018-1131-7 - further investigations in statics https://doi.org/10.1209/0295-5075/116/37002 https://doi.org/10.1103/PhysRevE.98.032139 and dynamics https://doi.org/10.1088/1751-8121/aa77e1 - including subdiffusive behavior https://doi.org/10.1209/0295-5075/117/30003 https://doi.org/10.21468/SciPostPhys.6.1.014

There are some (multifractal) generalizations of the Rosenzweig-Porter models with fat tailed distributions of off-diagonal elements: starting from Levy-Rosenzweig-Porter: https://doi.org/10.1088/1751-8121/aa77e1 - also mentioned above in the dynamics https://doi.org/10.1103/PhysRevB.103.104205 -Log-normal Rosenzweig-Porter: https://arxiv.org/abs/2002.02979 https://doi.org/10.1103/PhysRevResearch.2.043346 = [48] https://doi.org/10.21468/SciPostPhys.11.2.045 - including the subdiffusive dynamics

Even in short-range Floquet-driven systems one can observe multifractal phases: https://journals.aps.org/pre/abstract/10.1103/PhysRevE.81.066212 https://doi.org/10.1103/PhysRevE.97.010101 https://journals.aps.org/prb/abstract/10.1103/PhysRevB.103.184309 https://doi.org/10.21468/SciPostPhys.4.5.025 https://journals.aps.org/prb/abstract/10.1103/PhysRevB.93.104504 https://doi.org/10.1103/PhysRevB.106.L020201 https://scipost.org/SciPostPhys.12.3.082

In addition, in the correlated setting of the on-site disorder with short-range hopping, there is a whole bunch of works on Aubry-Andre model with p-wave superconducting pairing, showing a fractal phase. This wave has probably started with two works http://dx.doi.org/10.1103/PhysRevLett.110.176403 http://dx.doi.org/10.1103/PhysRevLett.110.146404, followed by the phase diagram calculation of the fractal phase in https://journals.aps.org/prb/abstract/10.1103/PhysRevB.93.104504 and now has quite a number of publications (please see the works citing the latter one). Please consider to cite some of the representative papers in your work.

d - It is rather hard to go back and forth in reading the numerical part of the manuscript as it refers to the analytical part quite heavily. Please consider to re-arrange the manuscript in such a way to make it readable without massive back-and-forth scrolling.

e - The same is true about the location of the numerical figures: please place them in the corresponding places, where you discuss them, but not a couple of pages before.

f - The usage of the notion of the inverse participation ratio in (31) and Fig. 12 is very confusing as it is related usually to the fractal dimension $D_2$. Please call $I_2$ in (31) the second moment in order to avoid this confusion.

  • validity: good
  • significance: good
  • originality: high
  • clarity: ok
  • formatting: reasonable
  • grammar: excellent

Author:  Ankita Chakrabarti  on 2023-09-22  [id 4000]

(in reply to Report 2 by Ivan Khaymovich on 2023-05-16)

Our response to this Report is contained in the attached file.

Attachment:

Reply_dzI4rrK.pdf

---

## Round 2 · Referee Report · Ivan Khaymovich (Referee 2) · 2023-9-22

Report

The authors have addressed all of my questions thoroughly and improved the manuscript.

I will comment only on a couple of more things:

Question 1. Yes, indeed, it is my fault in mentioning the Jordan-Wigner transformation for the loop-less graphs. It is the 1d structure which is important for that mapping.

Questions 2-4 were related to the difference between the Rosenzweig-Porter model, where the Thouless energy scales down as a power of the system size ($N$), and the short-range models (on the Cayley tree or RRG), where the Thouless energy remains finite or scales as a (power of the) diameter of the graph ($\sim \ln N$). But, as the authors focus on the ground-state physics, most of my questions seem to be irrelevant.

I have only the following minor reference issues:
a. [23, 27, 75, 97, 110] page numbers are missing
b. [30, 57, 68, 76, 78, 79, 85, 86, 93, 101, 108, 117] (doi) link is missing
c. Links to some arXiv are missing and the format is different: compare [81] to [25, 32, 99]
d. The format of references is heterogeneous in terms of the presence/absence of issue numbers.
Please correct them.

Overall I recommend the manuscript for the publication in SciPost Physics.

Requested changes

Please correct the following minor reference issues:
a. [23, 27, 75, 97, 110] page numbers are missing
b. [30, 57, 68, 76, 78, 79, 85, 86, 93, 101, 108, 117] (doi) link is missing
c. Links to some arXiv are missing and the format is different: compare [81] to [25, 32, 99]
d. The format of references is heterogeneous in terms of the presence/absence of issue numbers.

---

## Round 2 · Referee Report · Anonymous (Referee 1) · 2023-10-2

Report

I strongly recommend the publication of the revised preprint.

---

## Round 2 · Author Response

Dear Editor,
We are resubmitting our revised manuscript, titled "Traveling/non-traveling phase transition
and non-ergodic properties in the random transverse-field Ising model on the Cayley tree," for your
consideration for publication in SciPost Physics. We appreciate the time taken by you and the
reviewers in evaluating our work.
We are delighted that both reviewers found our manuscript interesting and valuable for the
community. Referee 1’s comments were particularly positive, strongly recommending its publication
in SciPost Physics.
Referee 2 provided a very detailed report, raising important and interesting questions regarding the analogy between the random transverse field Ising model we consider and the Anderson transition, both on the Cayley tree. In response, we have provided detailed explanations and made revisions to the manuscript to clarify these points. We have also followed their advice to enhance
the overall presentation of our manuscript. Furthermore, we have meticulously reviewed the bibliography.
To facilitate your review process, we have highlighted all changes made to the main text in
magenta within the manuscript.
We greatly appreciate your consideration of this revised manuscript and eagerly await your response.
Sincerely, the Authors.

---

## Round 2 · List of Changes

1. We have changed the abstract clarifying what "constant and algebraically vanishing boundary conditions" mean.
  2. We have added a sentence to the second paragraph of the introduction, discussing the substantial body of work describing non-ergodic delocalized/multifractal phases.
  3. We have included a paragraph in the introduction that discusses the Jordan-Wigner fermionization of the spin model on the Cayley tree in comparison to the 1D case.
  4. We have included two new paragraphs in the introduction that discuss the analogy with the traveling wave problem in the context of similar behaviors between the random transverse field Ising model and the Anderson transition on the Cayley tree.
  5. We have combined the previous sections 4 and 5 into a single section 4, where numerical results are discussed immediately after the corresponding theoretical predictions.
  6. We have reorganized the placement of the figures, positioning them as closely as possible to the corresponding discussions in the text.
  7. We have included a comment in the current Figure 5 (formerly Figure 3) about the bimodal shape of the distribution.
  8. We have inserted a paragraph at the end of Section 5, ’Non-ergodic phase,’ to address how our approach enables us to handle the intricate finite-size effects.
  9. In Section 6, we have replaced the term ’inverse participation ratio’ with ’second moment’ for I 2 in Eq. (31) and Fig. 12.
  10. In the conclusion, we have included two paragraphs to discuss the analogy between our cavity mean field approach and Anderson localization, devoting a paragraph to emphasize the limitations of this analogy .
  11. In the conclusion, we have added a paragraph discussing the differentiation between the Bethe lattice and the finite Cayley tree, as well as the intriguing perspective of Random Regular Graphs and small-world networks. We also discuss about the intricate finite-size effects in the context of the the non-ergodic ordered phase.
  12. In the conclusion, we have added a paragraph discussing comparison of the outcomes of our cavity mean field approach with other approaches.
  13. We have meticulously reviewed the bibliography. We have removed the citations to Ref. [30] when exclusively discussing the Cayley tree case. To facilitate the review process, we have highlighted all changes made to the main text in magenta within the manuscript.

---

## Editorial Decision

published